# Regulatory Frameworks and State-of-the-Art Decontamination Technologies for Recycled Polystyrene for Food Contact Applications

**DOI:** 10.3390/polym17050658

**Published:** 2025-02-28

**Authors:** Javiera Sepúlveda-Carter, José L. Moreno de Castro, Laura Marín, Paula Baños, Marcos Sánchez Rodríguez, Marina P. Arrieta

**Affiliations:** 1Departamento de Ingeniería Química Industrial y del Medio Ambiente, E.T.S. de Ingenieros Industriales, Universidad Politécnica de Madrid, C/José Gutiérrez Abascal 2, 28006 Madrid, Spain; javiera.sepulveda@upm.es; 2Grupo de Investigación: Polímeros, Caracterización y Aplicaciones (POLCA), 28006 Madrid, Spain; 3The Circular Lab, ECOEMBALAJES España, C/del Cardenal Marcelo Spínola 14, 28016 Madrid, Spain; j.moreno@ecoembes.com (J.L.M.d.C.); l.marin@ecoembes.com (L.M.); p.banos@ecoembes.com (P.B.)

**Keywords:** food contact materials, recycled polystyrene, recycling legislation, decontamination technologies

## Abstract

Recycling post-consumer plastics for food contact applications is crucial for the circular economy; however, it presents challenges due to potential contamination and regulatory requirements. This review outlines the current European and U.S. legislation governing recycled plastics in food contact materials (FCM). The European Food Safety Authority (EFSA) mandates the evaluation and authorization of recycling processes. This includes examining input/output flows, prioritizing the use of previously authorized FCM, and assessing decontamination efficiency through material-specific challenge tests. Additionally, it evaluates new installations intended to apply approved decontamination technologies. In contrast, the voluntary submission to the U.S. Food and Drug Administration (FDA) provides guidelines with general advice on methodologies and recommended parameters and challenge tests. Applications to the EFSA for non-PET materials, such as HDPE, PP, and PS, are reviewed, highlighting the challenges of each material. Recycled PS, with its lower diffusivity compared to polyolefins shows promise for food packaging, with potential as a next material approved for use in the European Union. Decontamination technologies for post-consumer PS are explored, including super-cleaning processes, solvent extraction, and industrial methods. The review emphasizes the need for multidisciplinary collaboration to address the uncertainties around potential contaminants and ensure the safety of recycled plastics for food contact applications.

## 1. Introduction

Today, the continued growth of environmental concerns has fostered the need to introduce plastics into the circular economy. In this context, different entities have proposed several goals to address the issue of plastic pollution, such as a 15% reduction in packaging in the European Union by 2040 per Member State per capita in comparison to 2018 [1]. In addition, the National Strategy to Prevent Plastic Pollution in the United States, published in November 2024, outlines six objectives aimed at eliminating the release of plastic waste from specific sectors into the environment by 2040 [2]. In 2023, 413.8 million tons of plastics were produced worldwide. However, only 8.8% are attributed to mechanical and chemical recycling, and a minor 7% to bio-based plastics [3]. These statistics denote that a substantial portion of plastics do not follow a circular path, and the purpose behind the implemented regulations is to reduce the impact of these materials at the end of their useful life. Although plastics are used in a wide range of industrial applications (including the agriculture field, the automotive industry, the building and construction sector, and for electronics applications, among others), the packaging sector is the largest consumer of plastics, reaching 39% in 2022 [4]. In this regard, the highest production of packaging is in the production of food packaging materials. Food packaging is mainly for short-term applications based on plastics derived from virgin fossil feedstocks, which generate a huge amount of plastic waste. Despite the increasing attention during the last decades on bio-based and biodegradable polymers for short-term applications (i.e., single use plastics, food packaging, etc.), polyethylene terephthalate (PET), polyolefins such as polyethylene (both high-density polyethylene, HDPE, and low-density polyethylene, LDPE), and polypropylene (PP), as well as polystyrene (PS) are still the most common plastics used for food contact applications [4,5]. However, as those polymers have short-term applications, the European Commission strategy for plastics proposes, as one of the key elements of the European Union’s plan (CEAP) for the reduction in plastic waste, enhancing both the cost-efficiency and quality of recycled plastics [6].

To increase the proportion of recycled plastics in the polymer production chain, one action contemplated is to implement mandatory recycled content requirements for products made from plastic, thus, reducing the need for virgin plastic and boosting the demand for recyclables [7]. Nevertheless, the introduction of recycled plastics into the plastic production chains should consider the final application of the plastic product made of polymeric formulation containing recycled plastics, not only for the final product overall performance, but also due to safety concerns.

In this regard, the importance of plastic in the packaging industry is mainly evident in the food sector [8], which is characterized by qualities that ensure the preservation, protection, and containment of food products at every stage, from production and distribution to sales and consumption, while providing relevant information of preservation and consumption to consumers, ultimately leading to the final disposal. Food-grade plastic packaging can be recovered by different means of recycling, with mechanical recycling being the most commonly used [9,10,11]. However, when intended for use as a Food Contact Material (FCM), complying with regulatory requirements presents additional challenges compared to the same virgin material. Besides considering the typical components present that might potentially migrate to food as presented in Figure 1 [12,13,14], additional sources of contamination are a major concern, the presence of non-intentionally added substances (NIAS) being one of the main factors to consider [15]. In this regard, Regulation (EC) 2022/1616 establishes that decontamination technologies related to recycled plastic for food purposes on the market are required to manufacture recycled plastic materials and/or articles containing recycled plastics, according to a suitable technology [7] or a novel technology (one awaiting authorization as suitable in accordance with the procedure described in chapter IV of the current legislation [7].

Consequently, strict food safety requirements are established by legislation for recycling processes originated from waste, addressing concerns such as the migration of substances that could impact human health, food quality, and microbiological safety, while also maintaining close control of the traceability of the materials and demands on recyclers, such as controlling contamination levels by carrying out compliance tests [16].

To obtain food-grade plastic products, different approaches are necessary depending on the type of recycling process. For instance, mechanical recycling does not alter their chemical structure, it typically consists of stages of collection, sorting, cleaning, and size reduction before further processing [17]. Current developments incorporate a decontamination stage to extract the greatest quantity of remaining contaminants and ensure that there is no health risk in exposing these materials to the conditions required in each standard to determine their suitability for food contact. On the other hand, chemical recycling aims to return the material to its starting monomer to further produce a polymer with the same characteristics as a virgin polymer. Chemical recycling refers to a process that transforms contaminated polymeric waste by breaking down its chemical structure to obtain substances that subsequently will be utilized either as products or as raw materials for the manufacturing of new items, excluding technologies used to generate energy [18].

Despite advances in circularity, chemical recycling has also incorporated alternative feedstocks, such as used cooking oil, to produce plastics [19], and reduce dependence on fossil resources. However, plastic waste management still relies mainly on landfill accumulation and incineration with energy recovery, where waste is converted into carbon dioxide, water, and heat, to be used in the production of electricity and/or heat.

Additionally, some less recognized recycling methods, such as those based on biological processes, have also gained popularity. Nevertheless, it is only applicable to certain polymers, i.e., compostable polymers. In this regard, biodegradable and compostable plastic products such as food packaging, biowaste bags, and cutlery strengthen organic recycling, mainly through industrial composting, which is an alternative waste management route that, instead of revalorizing the material into a new one, helps to increase waste management efficiency [20].

While biological degradation offers an alternative waste reduction route, its slow process limits large-scale applicability. In contrast, incineration provides a faster solution, yet it generates greenhouse gases and toxic byproducts like dioxins and heavy metals [21]. Due to the complexity of waste composition and the variability in the incineration process, municipal waste may not be fully incinerated. This incomplete combustion can generate microplastics and other secondary pollutants [22,23]. Ultimately, recycling is considered more environmentally beneficial than incineration, as it reduces CO_2_ emissions and environmental impacts more effectively [21].

Biobased and compostable plastics are not yet a definitive solution due to the limitations in their properties restricting their widespread use, especially in the food industry, which demands specific mechanical, thermal, optical, and barrier properties. Although biopolymers are a market that it is continuously growing [5,20], there is still extensive economic and infrastructure challenges to compete with traditional plastics used in food packaging. In fact, while the bioplastic production 2016 forecast for 2019 was 7.85 million tons of bioplastics (biodegradable and biobased non-degradable polymers), the actual production of bioplastics in 2019 reached only 2.11 million tons [20].

The most important aspect of technical European Food Safety Authority (EFSA) evaluation for post-consumer plastics intended to be in direct contact with food is essentially based on three main bounds: (i) concentrations of contaminants in the post-consumer input plastic materials from collection and pre-processing, (ii) parameters and cleaning efficiency of the decontamination technology applied in the recycling process, including a detailed challenge test and self-evaluation, and (iii) post-processing and intended use to ensure safety dietary exposure of residual and/or migrated contaminants to consumers [8,24]. Further comments on the European legal framework will be assessed in the following sections.

Post-consumer recycled polyethylene terephthalate (PCR-PET) is already approved by the EFSA to be used in the production of new food contact packaging products, primarily bottle-to-bottle recycling [25]. In fact, PET mineral water and soft drink bottles have been successfully recycled for the past two decades [26], leading to an optimized PET recycling process. As a result, nowadays, bottles with up to 100% PCR-PET content are approved and commercialized [25,27]. The favourable situation of PCR-PET with respect to other packaging polymers is mainly related to the fact that PET, the most well-known polymer used for beverage bottles, is practically the only material used in that application [25,28,29]. Therefore, huge amounts of one colour (blue, green, or brown) PET bottles can be easily separated from recycling streams, showing high efficiency in producing ready-to-process PCR-PET pellets [25,30]. Additionally, PET is a low-diffusive polymer; thus, low concentrations of substances are absorbed into the PET-based bottle during the first life of the beverage bottle, leading to low contamination levels [30], while PCR-PET mainly maintains many of its desired properties for food packaging purposes, such as high transparency, high mechanical strength and gas barrier performance, making it ideal for many foodstuffs and particularly for beverages [31].

Although PCR-PET is currently the only approved safe and regulatory-compliant material for manufacturing recycled food packaging, all post-consumer plastics should ultimately be recyclable to be introduced into the circular economy model. This principle also applies to biopolymers such as biodegradable polyesters, aiming to reduce the incorporation of virgin materials into the production chain and extend their life cycles before their final disposition in other streams such as compostable facilities. Nowadays, the presence of biopolymers in recycled streams can lead to the contamination of conventional recycled plastics, thereby hindering plastic recycling efforts [32].

Within this framework, polylactic acid (PLA), PHAs (polyhydroxyalkanoates) and their blends [33] are among the most promising and widely used biopolymers for food packaging. PLA, in particular, is currently the most commonly widely used bioplastic packaging sector due to its potential to address both commercial and environmental concerns in the market [20,34,35]. PLA is extensively applied as an alternative to petroleum-based plastics in food contact applications, such as beverage bottles, containers, overwrap, blister packs, lamination films, and foams [34] as a replacement mainly for PET [36], but also for PS [34,37] and polyolefins [34,38], among others.

In this context, PLA recycling has been studied as the best option for the end of life of the material, with composting in second place, especially when compared to landfill and/or incineration [39,40]. It has been observed that PLA can be mechanically recycled up to three times with low diminution on the melt-flow index and mechanical performance [41], being interesting for the mechanical recycling process, among other biopolyesters [42,43]. In this scenario, PHA homopolymers, such as poly(3- hydroxybutyrate) (PHB) and poly(3-hydroxybutyrate-*co*-3-hydroxyvalerate) (PHBV) [44], possess a narrow processing window with the melting temperature close to the degradation temperature being more suitable for chemical recycling than for mechanical recycling [44,45,46]. The bioplastic consumption is currently still low, and thus plastics products based on biopolyesters coming from recycled streams are very low. Therefore, the production of recycled bioplastics such as PCR-PLA or other PCR-bioplastics is still far away from the industrial sector, while they are more suitable to be recycled in a closed-loop, e.g., PLA-based products discarded from production lines can be reprocessed into pellets that are not derived from waste streams and have a well defined and traceable origin [42]. Therefore, the mechanical recycling of plastics for food contact materials beyond PET is currently focused mainly on traditional plastics. However, PLA and other biopolyesters (i.e., PHB, PHBV, etc.) are progressively being introduced in the food packaging field, and will shortly represent a considerable fraction of plastic waste [47]. Consequently, decontamination studies of post-consumer biopolyesters should be studied.

As polyolefins such as PE or PP are high-diffusive polymers that possess lower barrier performance than PET or PS, the next promising candidate to be potentially considered as PCR plastic for food contact proposes is PS [30,48]. PS’s unique characteristics, such as its high thermal stability and sorting efficiency in waste recovery systems over other plastics [49], ease of processing, excellent mechanical properties, low water vapour transmission, and low cost, has led to its wide use in food packaging [48]. Meanwhile, it is a low-diffusive polymer like PET, which positions PCR-PS as a promising candidate for food contact applications with particular interest as a functional barrier (FB) [30,48].

Nonetheless, the separation of polystyrene (PS) from other plastics still poses significant challenges due to the lack of dedicated collection systems within plastic waste streams, hindering efficient management. Its physical similarities to polymers such as Acrylonitrile Butadiene Styrene (ABS) and polypropylene (PP) further complicate identification and sorting through conventional technologies [50,51]. Industrially, sink–float separation, a widely used wet method, exploits differences in plastic densities to facilitate separation. Additionally, mechanical sorting techniques such as X-ray detection, near-infrared (NIR) spectroscopy, and VIS (colour analysis by camera or spectro-colourimeter) are commonly employed to analyze polymer composition [50,52]. However, NIR has limitations in detecting PS due to its weak spectral absorption and similarity to other polymers, often requiring complementary technologies for accurate separation [53].

Recognizing the importance of short-term solutions in recycling, this review primarily focuses on analyzing the current European and American regulations overseeing the use of recycled materials in charge of the European Food Safety Authority (EFSA) and the Food and Drugs Administration (FDA), respectively. Furthermore, it assesses the changes in legislation conducted by each regulatory body and highlights the key differences between the two approaches, concluding with an overview of the status of decontamination studies applied particularly to polystyrene (PS) for its potential to become the next plastic to be approved for use at the European level as a post-consumer recycled material.

## 2. Legal Framework

Different organisms are designated in each region to determine and guarantee that materials or objects intended to come into contact with food meet safety requirements, evaluating and granting authorization of use. Two of the most relevant global regulations in the Western world to be analyzed are the European Union legislation, in charge of the European Commission through the European Food Safety Authority (EFSA), and the Food and Drugs Administration (FDA) in the USA.

### 2.1. European Legislation

As part of the European Union’s (EU) long-term strategy to reach climate neutrality by 2050, in 2018, it published its plastic strategy in a circular economy [6], along with the Circular Economy Action Plan (CEAP) under the European Green Deal [1]. Aiming to decrease plastic residues, the plan states that all packaging should be designed to be recyclable, reusable, or compostable in a cost-efficient manner by 2030, with specific goals for incorporating recycled material into new packaging. The basis for the use of plastics in contact with food relies on Regulation (EC) No 1935/2004, which states that there can be no transfer of its components to food that could put human health at risk or alter the organoleptic characteristics and appearance of the product. Likewise, processing must be performed under good manufacturing practises according to Regulation (EC) No 2023/2006 and comply with the specifications detailed in Regulation (EU) No 10/2011 for plastics materials and articles intended to come into contact with food.

The Packaging and Packaging Waste Directive 94/62/EC of 1994 (PPWD) has been the basis for packaging management throughout their life cycle. The European Commission initiated a review of the 1994 PPWD in 2022 under the procedure 2022/0396/COD to establish goals that promotes packaging reduction including the use of recycled plastic, therefore, aiming to have higher quality recyclates to replace virgin materials [54]. In April 2024, the new Packaging and Packaging Waste Regulation 2025/40 [55], now PPWR, was approved at a first reading by the European Parliament, that derogates Directive 94/62/CE and modifies Regulation (UE) 2019/1020 and Directive (UE) 2019/904 [56]. After the favourable position of the just-elected European Parliament to the corrigendum in November 2024, and the approval of the Council on 16 December 2024, the legislative text was adopted and entered into force on 11 February 2025 [57]. The new text applies restrictions on some packaging formats, for example, banning certain single-use plastic packaging from 1 January 2030, and encourages reuse and refill systems.

The new framework approved proposes a more ambitious focus, in line with the European Green Deal, to chart a course that ensures all packaging would be recyclable or reusable in a feasible manner by 2030. It also centres on preventing and reducing the adverse effects of packaging and packaging residues in the environment and human health, contributing towards a circular economy. It establishes some more specific criteria for the practical evaluation of recyclability requirements, implementing a classification system (A, B, C) that evaluates the level of ease and economic viability of recycling packaging. Category A indicates highly recyclable packaging, in an economic and technical manner, Category B, corresponds to packaging that is recyclable with certain limitations or additional processes to ensure effective recycling, and finally, Category C, relates to packaging with lower recyclability, whose commercialization in the European Union would be forbidden from 2038.

A national contribution to non-recycled plastic packaging waste was established in Council Decision (EU, Euratom) 2020/2053 of 14 December 2020 establishing key financial measures to ensure responsible financial management. The decision maintained traditional revenue sources and temporarily raised the EU budget ceiling to provide more financial flexibility. It also introduced a uniform call rate of €0.80/kg on non-recycled plastic packaging waste generated in each Member State, aiming to promote recycling and reduce environmental impact [58].

Firstly, plastic packaging producers, through the Plastics Europe association, proposed a mandatory target of 30% inclusion of recycled content for all plastic packaging [59]. Currently, parts of these goals have been adopted in different countries’ legislations [60]. Moreover, all countries associated with the European Union are required to comply with the recent Regulation (EC) No 2022/1616 to market a recycled material in contact with food, related particularly to recycled plastic materials and objects intended to come into food contact [7]. This regulation maintains the base enforced by its predecessor, Regulation (EC) No 282/2008 [61], and updates both concepts and procedures.

The approval process for recycled plastic materials for food contact involves conducting a comprehensive characterization of the materials and comparing the generated data and information using different analytical approaches. This suggests the need for an in-depth analysis of the recycled plastic materials necessary to ensure their safety and suitability in food contact. In 2008, the European Food Safety Authority (EFSA) published guidance assessing the application procedures to submit a safety assessment of a recycled plastic technology, with information on the technical and administrative aspects [15], followed by guidance on transparency in the scientific aspects of risk assessment [62]. In addition, a complete overview of the different aspects of the evaluation approach of recycled PET was published in 2011 [63], and updated in 2024 [24] to modify existing authorisations. The scientific evaluation approach is based on applying the decontamination effectiveness of a recycling process, determined through a challenge test using surrogate contaminants, to a baseline contamination level for post-consumer PET established at 3 mg from potential misuse in 1 kg PET. Each surrogate is compared to its conservative modelled concentration, equivalent to more than 0.0025 μg/kg body weight per day of dietary exposure.

In 2021, the “Administrative guidance for the preparation of applications on recycling processes to produce recycled plastics intended to be used for manufacture of materials and articles in contact with food” was published [64]. Detailed guidelines according to the 2022 regulation have since been updated [7] through administrative guidance for applications on individual recycling processes [8].

Under the previous regulation, any decontamination technology that proposed incorporating recycled material in contact with food into the market required going through an application process that considers an evaluation by the EFSA, in particular, the Panel on Food Contact Materials, Enzymes, Flavourings, and Processing Aids (CEF), and subsequent authorization issued by the European Commission (EC). This panorama resulted in PET emerging as the only material from conventional post-consumer mechanical recycling (PCR), capable of meeting the specifications of all the aforementioned regulations in its process, enabling its current commercialization and use as a recycled material in contact with food. The current status of PET is a consequence of its physical and chemical characteristics, primarily its application in bottles that facilitate their efficient identification, collection, and sorting during the process, which, along with the advancements in research, contributed to a decrease in non-food grade plastics contamination [65,66]. Regarding materials other than PET, only materials coming from post-industrial mechanical recycling (PIR), collected in a closed loop or a highly traceable source, were authorized before the change in regulation, as strict controls are enforced over the decontamination process, and status is maintained with the differentiation in technology classification.

A classification of recycling linked to the techniques referred to, correlates with the origin of the materials used to manufacture a product [67]. Closed-loop recycling involves products made from materials that have been recovered from the same process by which they were originally manufactured, ensuring that the materials used maintain their properties. In contrast, open-loop recycling involves products made from different materials than those they were originally manufactured from, frequently associated with the utilization of materials from unidentified origins, which poses challenges in regulating the input material employed.

The current EU Regulation 2022/1616 on recycled FCM designates two main classifications: suitable recycling technologies and novel technologies. In this regard, suitable recycling technologies are capable of recycling waste into recycled plastic materials and articles that comply with Article 3 of Regulation (EC) No 1935/2004 and are also free from microbiological hazards. Therefore, they have proven to generate harmless waste through the process, an individual evaluation being necessary in the last case to ensure compliance with the criteria determined by the developer. This establishes a public register for the traceability of recyclers, recycling installations and processes. Currently, there are two technologies listed as suitable: (i) products in a closed and controlled loop recycling, and (ii) mechanical recycling of PET under specific conditions of input and use. For technologies considered as novel, articles 10–16 of Regulation 1616/2022 apply, and they must undergo a more exhaustive evaluation and authorization before being considered as suitable. Therefore, for these technologies, the developer shall submit to the European Commission extensive reasoning, scientific evidence, and studies demonstrating that the novel technology can be manufactured by complying with recycled plastic FCM normative, with microbiological safety, including a characterization of contaminant levels in the plastic input and the recycled product, a determination of the decontamination efficiency (at pilot or industrial level), migration tests, and reasoning for meeting those requirements [7].

The promotion of the development of novel technologies by temporarily allowing the use of these materials in the market subject to strict conditions, helps to gather data and minimize uncertainties about their characterization, adequation, and special requirements. In this way, recyclers who implement a decontamination process shall monitor the process of the decontamination installation under their control, including contamination of the input material (identification and quantification of contaminants), and maintain constant communication with the developer of such technology. A monitoring report shall be published by the developer every 6 months on their website based on the information from all the installations. After at least four consecutive reports, the developer could apply to the EC to initiate an assessment that allows them to put the recycled plastic materials on the market. The Commission is also able to initiate this process on its own initiative. If there is sufficient information, the submitted data will be evaluated by the EFSA and a specific opinion with the relevant requirements within 1 year of data received should be published as the outcome. The evaluation might be postponed for up to two years to gather more information.

In July 2024, Regulation (EU) 2024/1781 of the European Parliament and of the Council on 13 June 2024 came into force, establishing a framework for the setting of eco-design requirements for sustainable products, amending Directive (EU) 2020/1828 and Regulation (EU) 2023/1542 and repealing Directive 2009/125/EC. It establishes a framework for eco-design requirements applicable to sustainable products, expanding its scope beyond energy-related products. It introduces the “digital product passport” to improve traceability and sustainability information and prohibits the destruction of unsold consumer products [68].

The Commission is responsible for making the final decision on changing a certain technology’s status and deeming it suitable. Moreover, the regulation incorporates new concepts to its scope like chemical recycling, making a differentiation in the removal of contaminants from substances and mixtures falling under the term purification as a form of decontamination. Functional barriers that may fully contain such contaminants are introduced as novel technology in the pursuit of defining specific requirements that verify how effective the use of recycled materials to prevent migration in the long term can be [7]. It also establishes a public register for the traceability of recyclers, recycling installations, and processes.

#### Non-PET Recycling Processes Evaluation for EFSA Authorization

As was already mentioned, PCR non-PET plastics should be included soon in food contact materials manufacturing. Therefore, applications submitted to the European Commission for the authorization of the use of decontamination technologies that can produce materials safe for food contact under specific conditions were reviewed. The data examined includes all open applications published under previous recycling Regulation (EC) No 282/2008 in the EFSA Journal since 2008 and applications submitted to Open EFSA between the website’s launch in 2021 and November 2023, totalling 182. Out of these, only 26 applications involved materials other than PET [69]. Applications in the intake phase that did not proceed through to the evaluation process and did not clearly identify the material to be treated (or neither can be deduced) were not considered for the purpose of this review. The specific details of each application, including registration of the material to recycle, the status of the submission, the recycling process information (EFSA question ID, name, decontamination technology), the input and output product involved, and the applicant and country that initiated the development, are provided in Table 1.

High-density polyethylene (HDPE) and polypropylene (PP) head the applications, while PS is in third place due to the efforts of Styrenics Circular Solutions (SCS) [70], a joint industry initiative aimed at significantly enhancing the circularity of styrenics, with a particular focus on the use of post-consumer waste, and where Ineos Styrolutions is member. However, by focusing on the latest plastic recycling regulations, applications for technologies submitted under previous Regulation (EC) No 282/2008 were not considered suitable, and product loops that were in a closed and controlled chain were deemed terminated [7].

**Table 1 polymers-17-00658-t001:** Applications under EFSA evaluation for recycled polymers different to PET, including material evaluated, status of the evaluation, details of the recycling process, decontamination technology, input and output materials to be used, applicant, and country.

Material ^i^	ID—Process	Type ^ii^	Status ^iii^	Specific Conditions of Use for the Application		Applicant ^iv^ Country
	Decontamination Technology			Input	Product	
PP	EFSA-Q-2009-00682—**PP crate CHEP** [71]	CC	A	Crates	Crates for fruits, vegetables, pre-packed meat	CHEP UK Ltd.—United Kingdom (Netherlands ^vi^)
HDPE	EFSA-Q-2009-00961—**Biffa Polymers** EREMA advanced technology [72]	S-OL	A	Bottles, mainly from milk (from exclusive providers)	≤30% + virgin PP for single-use trays for raw fruits, vegetables, animal products, mushrooms	Biffa Polymers Limited United Kingdom (U.K.)
HDPE	EFSA-Q-2010-00020—***CLRrHDPE*** EREMA advanced technology [73]	OL	R		≤50% + virgin HDPE for refrigerated juice and milk bottles	Closed Loop Recycling Limited—U.K.
PP HDPE	EFSA-Q-2010-00021—**CO.N.I.P. (National Plastic Packaging Consortium)** [74]	CC	A	Crates	Crates for whole fruits and vegetables	CO.N.I.P. Italy
PP HDPE	EFSA-Q-2010-00068—**Schoeller Arca Systems** [75]	CC	A	Crates	Crates for meat, whole fruits, and vegetables	Schoeller Arca Systems B.V.—Netherlands
PP	EFSA-Q-2010-00104—**Petra Polimeri** [76]	CC * S-OL **	A	Insert trays * or trays **	≤30% + virgin PP. Trays or insert trays for whole fruits and vegetables.	Petra Polimeri S.R.I.—Italy
PP	EFSA-Q-2010-00892—**INTERSEROH Step 1** [77] EFSA-Q-2010-00951—**INTERSEROH Step 2** [78]	CC CC	A A	Crates	Crates for whole fruits and vegetables	INTERSEROH Dienstleistungs GmbH– Germany (Netherlands ^vi^)
PP HDPE	EFSA-Q-2015-00444—**Pokas Arcadian Recycle Ltd.** Pokas Arcadian, batch process [79]	MIX	R	(1) Crates, scraps, (2) packaging from a closed collection system, or (3) approved packaging from recycling system	Crates, packaging, or as a functional barrier for whole fruits and vegetables	Pokas Arcadian Recycle Ltd. Greece
PP HDPE	EFSA-Q-2016-00486—**Morssinkhof Plastics** [80]	MIX	A *	Crates, boxes, trays, pallets, and containers (* Excluded regrind from external recyclers)	Crates for dry food, fruits, vegetables, prepacked, and unpacked meat	Morssinkhof Plastics Lichtenvoorde B.V. Netherlands
PCTG	EFSA-Q-2019-00016—**Green Loop System** [81]	CC	A	Plates	Plates for contact with aqueous, acidic, and fatty food	Mälarplast AB Switzerland
HDPE	EFSA-Q-2019-00296—**Starlinger recoSTAR HDPE FC 1—PET2PET** Starlinger recoSTAR HDPE (FC 1) [82]	OL	R	Bottles closures	Bottle closures for mineral water and beverages for long-term storage at room temperature	PET to PET Recycling Österreich GmbH Austria
PP HDPE	EFSA-Q-2020-00458—**Loop Polymers** Internal + optional EREMA refresher stage [83]	CC	R	Unused offcuts and scraps carrying print coatings, inks, or adhesives	Food contact packaging	Loop Polymers Ltd.—United Kingdom (Ireland ^vi^)
PP, PET, SAN, ABS	EFSA-Q-2020-00231—**deSter** [84]	CC	R	Catering tableware from airline on-board services	Articles for the same on-board services	deSter BVBA Belgium
HDPE	**Schwarz Produktion MEG Weißenfels**	ND	NV	Crates, pallets, and bottles closures from PET recycling process (Drescher and Kauertz, 2023)	Crates, pallets and bottles closures (Drescher and Kauertz, 2023)	MEG Weißenfels GmbH & Co. KG Germany
HDPE	EFSA-Q-2020-00511—Erema HDPE regrind pro plus refresher [85]	ND	T			
HDPE	EFSA-Q-2020-00772—**Kunststof Recycling Nederland (KRN)** [86]	CC	A	Box pallets for meat packaging.	Box pallets for refrigerated or frozen, packed, or unpacked meat.	KRN Netherlands
PS	**Styrenics Circular Solutions** EFSA-Q-2021-00151—Next Generation Recyclingmaschinen (NGR) [87] EFSA-Q-2022-00039—Gneuss 3, Multi Rotation System (MRS) [88,89]	OL	OG	Post-consumer food-packing waste from European collections systems [89,90].	Yoghurt pots, hot and cold beverage containers, and food trays [90,91,92].	Styrenics Circular Solutions Belgium
PS		OL	T			
PP HDPE	EFSA-Q-2021-00190—**THEES** **Kunststoffverarbeitung** [93]	CC	T	Crates	Not indicated	THEES GmbH Germany
PP HDPE	EFSA-Q-2021-00294 **Cajas y Palets en una Economía Circular (CAPEC)** [94]	CC	A	Crates	Crates for whole fruits and vegetables	CAPEC Spain
PP HDPE	EFSA-Q-2021-00336—**LOGIFRUIT** [95]	CC	A	Crates	Crates for fruits and vegetables	LOGIFRUIT S.L. Spain
HDPE	**Craemer** EFSA-Q-2021-00416—Leistritz extruder [96] EFSA-Q-2021-00411—Erema HDPE regrind pro plus refresher [97]	OL * CC **	T T	Post-consumer closures *, crates, and pallets **	Crates and pallets	Firma Craemer GmbH Germany
HDPE	EFSA-Q-2021-00783—**AST Recycling** **& Rekonditionierung** [98]	CC	T	Canisters [99,100]	Canisters [99,100]	AST Recycling & Reconditioning GmbH & Co. KG—Germany
PS	EFSA-Q-2022-00202—Ineos-styrolution Internal, twin screw degassing extrusion [101]	T	OL	Post-consumer	Yoghurt cups, food trays	INEOS Styrolution Switzerland S.A.—Switzerland (Germany ^vi^)

^i^ HDPE: High-density polyethylene; PP: Polypropylene; PCTG: Polycyclohexylene dimethylene terephthalate glycol-modified; PET: Polyethylene terephthalate; SAN: Styrene acrylonitrile resin; ABS: Acrylonitrile butadiene styrene; PS: Polystyrene. ^ii^ CC: Closed and controlled loop; OL: Open loop; MIX: CC and OL; ND: Not determined. ^iii^ A: Accepted; R: Rejected; NV: Not valid; OG: Ongoing evaluation; T: Terminated. ^iv^ Does not consider the intermediate, only the entity that presents the initiative for the recycling process. ^v^ If not indicated, it means an internal procedure implemented. ^vi^ Member state where the EFSA evaluation is requested on behalf of the applicant. *; ** indicate specific conditions that apply only to terms marked with the **same symbol** within an **application** and are **not related across different applications**.

Thus, the recycling processes that are not determined as a safety concern are exclusively from a closed loop system [102]. In the few cases where applications were approved with a certain degree of open-loop involved, very strict specified terms are indicated, in terms of (1) input material involved, in general, with a proved traceability and control, (2) considering an input from previous approved food contact materials, (3) safeguarding the input stream, and (4) passing by an exhaustive analysis not only of the original plastic products but also including suppliers, even in the sorting stage, where they are submitted to audit to be approved. One of the measurements considered to have approval in applications is the promotion of less likely migration scenarios, as products requiring low temperatures or short contact times [102,103,104] have specific limits in the incorporation in blends with virgin, for example, the use of up to 30% recycled material mixed with virgin material. Meanwhile, cases with a close and controlled input are used up to 100% [105,106].

A case example of this scenario is the Biffa Polymers and CLRrHDPE proposal in 2015 [107], which utilized HDPE post-consumer bottles mainly in contact with milk as the input material; nevertheless, uncertainties found in the data provided for their use in milk and fruit juice bottles in conjunction with trays for animal products, leading to the application only being authorized for the use of recycled material in trays intended to be in contact with whole fruits, vegetables, and mushrooms, utilized for transport, storage, and display. An important factor in the decision was the research on input materials, with the suppliers and the manual and automatic sorting process reaching at least a 99% stream of recycled material coming from food-grade post-consumer materials. However, it is worth noting that private agreements with third parties involved are not considered sufficient reassurance to decrease any contamination of the input plastic [104]. The case provides a more comprehensive scientific viewpoint on the evaluation of HDPE procedures by the authorities.

On the other hand, the main challenge highlighted by the EFSA in the reviewed and published applications lies in the need to analyze the input material rather than focusing solely on the recycled material to identify the chemicals involved and properly define contamination scenarios and potential surrogates. Additionally, the study of diffusion coefficients at low temperatures is emphasized as a key factor in improving migration models. In general, migration tests are carried out according to previous methodologies, through the total immersion of materials recycled once and five times, with three replicated tests performed. Overall migration and specific migration of substances with a specific migration limit (SML) in compliance with Regulation EU No. 10/2011 into critical conditions were selected.

For instance, Vera et al. have studied the migration of volatile substances from post-consumer mechanically recycled HDPE-based milk bottles and observed that both degradation products (coming from antioxidant compounds and several residues from cleaning products, detergents, and flavouring agents) and NIAS were found after the exposure of the materials to a fatty food simulant (EtOH 50% *v*/*v*). They performed a risk assessment and found that the compound 1-dodecene exceeded its specific migration limit (SML) [108].

Regarding the cleaning to remove adhesives, coatings, ink systems, and dirt, among others, some common procedures mentioned grinding and/or incorporating a centrifugation or sieve stage to remove small particles, followed by a washing step [104,109] with water (in general hot) and chemical agents, such as detergents (alkaline [110], acidic [111], or neutral [105]) and disinfectant [105,110] with posterior water rinsing.

Conclusively, drying to proceed to the final production of the recycled material, granulated by means of injection moulding or extrusion under vacuum, favours the decontamination of volatiles and can help to discount the possibility of contamination by microorganisms due to the high temperatures [104,112]. Decontamination efficiency must be estimated based on a method with an analytical specificity for the different components and a known tolerance, to prove if an appropriate challenge test representative of the industrial process with surrogate contaminants of different molecular weights and polarities, or, if any adequate evidence is necessary, this might be supported by migration models [104,105,111].

Supercritical CO_2_ extraction is a sustainable and environmentally friendly method for purifying polyolefins, with the added benefit of eliminating expensive post-treatment steps required in solvent-based methods to remove residual solvents. It is also safer and easier to handle compared to organic solvents, in terms of waste management at an industrial scale [113]. However, the requirement for operating at pressures above 70 bars poses a significant challenge, as the high costs associated with the equipment and operation could make the process economically unfeasible for large-scale applications [114].

The contaminant analysis should include both polar and non-polar compounds, with molecular weights up to 1000 Da [110]; equally important is the evaluation of health risk, considering a threshold of toxicological concern (TTC) under 0.0025 μg/kg of body weight per day for unknown contaminants represents a negligible risk to the consumer [63]. Printing inks that can be found in packaging can be a source of concern since they are not regulated at the European level; therefore, they can contain substances that are not authorized for food contact. Considering that materials such as PE and PP have weak barrier properties, low molecular mass compounds are likely to diffuse into the plastic.

Management of the whole process (collection, sorting, recycling, distribution) through a quality assurance system (QAS) is critical [112], as well as a product design that minimizes the likelihood of contamination through misuse. Conditions in the mentioned cases outside close loop processes are very difficult to maintain at an extended scale, where more variables and external parties need to be considered, implying an undetermined and inconstant input composition.

### 2.2. Food and Drug Administration

The U.S. Food and Drug Administration (FDA) is authorized by a set of United States (U.S.) laws in the Federal Food, Drug, and Cosmetic Act (FDCA) to ensure the safety and intended application suitability, supervising, and overseeing of the production, sale, and distribution of food, drugs, medical devices, and cosmetics, in accordance with good manufacturing practises [115].

Title 21 of the Code of Federal Regulations (CFR) for Food and Drugs, Chapter I, Subchapter B: Food for Human Consumption oversees the use of food-contact substances, including additives and packaging materials. In parts 174 to 179, specific regulations on indirect food additives are included, with section 21 CFR 174.5 stating that any substance used in contact with food must be of a purity suitable for its intended use, based on the evaluation of recycling processed with the aim of food packaging applications.

Requesting and obtaining an authorisation to use recycled material for food-contact packaging in the United States involves several key steps. This process typically leads to the issuance of a No Objection Letter (NOL), a document from the FDA confirming that a recycling process meets safety standards for food-contact use. However, this procedure is not mandated by any legal or regulatory requirements and remains voluntary [116]. Nevertheless, for recycled plastic to be legally used in food contact applications, it must meet the safety and purity specifications established in the above mentioned regulations [117]. Therefore, the FDA regulates food contact materials individually, on a substance-by-substance basis, and each is individually subject to one of the following statuses [118]:-The component is listed in Title 21 of the U.S. Code of Federal Regulations (21 CFR), specifically the database that contains an inventory of Food Contact Substances (FCS) authorized for uses as an appropriately regulated indirect additive in contact with foods. This database includes components of materials used in the manufacturing process, packaging, transport, or food containment, but only if they are not intended to have any technical effect on the food; it is updated yearly.-Meets the criteria for ‘generally recognized as safe’ (GRAS) status, which includes, but is not limited to, a GRAS regulation (after passing a review process by the FDA) or GRAS notice (after approving a voluntary submission) [119].-Owns a prior sanction letter, granted by the FDA or USDA before 1958, expressing no opposition towards the utilization of a particular substance for a specific purpose.-The FCS manufacturer holds an effective Food Contact Substance Notification (FCN), that specifically covers the authorized uses and conditions of use. This regulation applies to manufacturers who have notified the FDA of their intention to utilize an FDA-approved FCS by submitting an FCN. The regulatory authority must conduct a thorough review of the scientific data within 120 days. If no objection is raised during this period, the FCM is considered ‘effective’ [120].-A Threshold of Regulation (TOR) Exemption has been issued for the component; therefore, it does not require a Food Contact Substance Notification. This exemption addresses regulatory matters, as the substance has already been evaluated and deemed safe at exposure levels below the regulatory threshold (0.5 ppb) when migrating from packaging into food [121].

An additional criterion includes the option to request a Food Contact Formulation (FCF) Notification, to ensure that the components of a specific food contact material meet compliance requirements. However, it is important to note that this notification only serves regulatory purposes and is not intended for conducting a new safety assessment, just to ensure all substances used in the formulation are already authorized [122].

Informal categorizations are used to classify food additives [123]. Direct food additives are substances that have a technical impact within food. Secondary direct additives serve a purpose during food processing but do not remain in the food. Indirect food additives are substances used in food contact materials that may migrate into food but are not added intentionally or to have a technical effect in or on the food. 

To ensure safety, every authorization for food-contact substances must have three essential components. First, it should identify the substance. Second, it should cover substance specifications enclosing its purity or physical properties, along with any limitations on its use. Finally, the manufacturer, as the responsible party, should provide a letter of guarantee to their customers, as certification that the product is suitable for the intended food-contact use [118]. The assessment by the United States authority places significant emphasis on the migrating substances nature and presence below harmful levels in terms of dietary exposure of consumers alongside their potential capacity for accumulation.

The authorization of food additives before they are marketed to the public usually goes through an FCN. The scientific assessment includes determining if a provable amount of the FCS migrated to food, based on testing and toxicological data to ensure no risk to the consumer or the environment under the National Environmental Policy Act [124]. Several guidance documents [125] have been developed by the FDA to assist the industry with submissions for food ingredients and packaging, including scientific guidance documents on chemistry, microbiology, and toxicology.

Notifications for food contact are not a request to the FDA, but instead regard the intention of the company to introduce a new FCS to the market or use it for a new application. The authority presents an acknowledgment letter and, if all the information is correctly in place and without any objection to the notification in the time established, the FCN becomes effective, concluding the process with a final letter expressing the details of the process reviewed [124].

The evaluation of FCM is conducted on a voluntary basis, and if the information on the recycling process provided demonstrates its safety for food contact, a favourable opinion letter is issued. The FDA’s main concerns involve the incorporation of food-contact post-consumer recycled (PCR) materials into the final product and additives to the process, that in both cases might not comply with current regulations for food contact applications, as well as the likely persistence of contaminants coming from the input sources [116,126]. In order to address said matters, the FDA prepared a guide to facilitate the industry comprehension regarding chemical aspects in the use of recycled plastics for food packaging [117], simultaneously delivering informal advice on the suitability of each recycling process.

The guide describes the approach recommended for the evaluation of efficacy, adequacy, and the gathering of data for a recycling process in the removal of chemical contaminants, by presenting distinctly established assumptions and methods to approach the evaluation by each entity. Three fundamental types of recycling processes are defined as a basis for this assessment:-**Primary recycling**—Pre-consumer scraps from food contact in the industry. Product from a closed-loop chain, considered low risk; however, if more than one manufacturer is involve in the material to be treated, further evaluation is required to prove it remains non-hazardous to the consumer.-**Secondary recycling**—Post-consumer plastic packaging materials in mechanical recycling, maintaining the nature of the polymer. Additives, antioxidants, processing aids, and other substances involved must be reduced to levels of no concern and comply with current regulations. The assessment conducted by the FDA encompasses a range of factors, including the implementation of controls on source materials, effective sorting procedures, and limitations on the application of these materials in specific contexts and food types.-**Tertiary recycling**—Post-consumer plastic packaging through chemical recycling. Polymers or monomers can undergo various stages of purification in order to be repolymerized and can possibly be combined with virgin materials. The entire process may include multiple stages of purification, including washing, distillation, crystallization, and additional chemical reactions.

The FDA’s method for assessing the exposure of consumers to contaminant evaluates the probability of long-term, cumulative exposure to low-level contaminants, rather than evaluating each compound individually. As a result, levels of contaminants potentially migrating from recycled plastics that are considered to be of negligible risk are no more than 1.5 micrograms per person per day as the estimated daily intake (EDI), equivalent to 0.5 parts per billion (ppb) of dietary concentration (DC).

Calculating the limit amount of a contaminant remaining in the process depends on each polymer according to its thickness and density; recommendations made by the authority indicate conservative assumptions: polymers manufactured for 100% recycled material are intended for contact with all type of foods (aqueous, acidic, alcoholic, fatty food), and a default consumption factor of 0.05 for any recycled plastic, based on the PET latest market data, making it the plastic with the highest recycling rate.

Under those parameters, for 0,50 mm thickness packaging, the maximum level for a contaminant trace in PET (density:1.4 g/cm^3^) results in 220 µg/kg, 300 µg/kg for polystyrene (PS) (density 1.05 g/cm^3^), 200 µg/kg for polyvinyl chloride (PVC) (density 1.58 g/cm^3^), and 320 µg/kg generally for polyolefins (density 0.965 g/cm^3^).

The primary method for evaluating the effectiveness of contaminant removal follows a similar path to that of the European authority, the EFSA. This entails conducting a “challenge” test, in which a material is exposed to a mixture, or “cocktail”, of specific substances or surrogates, using a solvent such as hexane. Virgin material, in the form of flakes or packaging, is then immersed or filled in the mixture for a period of two weeks at 40 °C, with constant agitation. After simulating the contaminated material, the polymer is drained and rinsed, and is then subjected to the recycling process, where it is assumed that all input is contaminated.

The FDA suggests that when using recycled plastic in food packaging, a protective barrier, such as a layer of virgin polymer or aluminum, can prevent the migration of potential contaminants from the recycled plastic to the food. This barrier should have a specific minimum thickness that depends on the temperature of use, ensuring that migration levels remain safe and within negligible risk limits for consumers [117].

The focus on safety in recycled plastics use aligns with national efforts to build a circular economy, as demonstrated by initiatives led by public entities. “Towards building a circular economy”, announced in 2015 by the U.S. Environmental Protection Agency (EPA) in collaboration with the U.S. Department of Agriculture (USDA), set two goals to be achieved by 2030: reducing food loss and waste by 50% and increasing recycling rates by 50% [127]. In 2021, the EPA published “National Recycling Strategy: Part One of a Series on Building a Circular Economy for All”, aiming to strengthen the municipal solid waste recycling system [128]. Focusing specifically on plastics, in June of 2024 the EPA, authorized by the Save Our Seas 2.0 Act, announced funding opportunities for the Solid Waste Infrastructure for Recycling (SWIFR) programme [129], which allowed the release of the “National Strategy to Prevent Plastic Pollution: Part Three of a Series on Building a Circular Economy for All”.

Different United States entities, including business academia, industry, non-governmental organizations, federal, tribal, state, local and territorial governments, and consumers, are encouraged to take both voluntary and regulatory actions to completely prevent plastic waste from being released into the environment from both land-based and marine sources by 2040.

To address this issue, six objectives were established, providing opportunities for action. Among these key objectives was an emphasis on identifying alternative materials, products, and systems that minimize impacts on human health and the environment. Specifically, these goals aim to reduce the manufacture and consumption of single-use plastics, improve composting systems, develop the capacity to reuse materials, and advocate for the implementation of a national extended producer responsibility (EPR) framework [130].

#### Non-PET Recycling Processes Evaluation for FDA Authorization

A total of 361 No Objection Letters (NOLs) were issued from 1990 to 2024 for recycling processes to produce plastics intended for use with food; 128 are associated with non-PET materials, and 74 of them were granted in the last 5 years. Figure 2 presents a division of the endorsements received by the type of non-PET plastic, with PE leading the processes, particularly with HDPE, followed by PP and closely behind, PS. Publicly open data for the evaluations in this case are not available.

Although Urethane-Acrylate, Hydrogenated Carbon, Epoxy, acrylic and acrylic-based polymers, Dimethyl terephthalate (DMT), and Ethylene Glycol (EG) correspond to the non-PET category, they are approved for use in the PET recycling process, which is why they were not considered for this study.

The use of chemical or third recycling processes as a relatively new technology holds only two approved applications. The first, which dates back to 1996, involves PEN (Poly(oxy-1,2-ethanediyloxycarbonyl-2,6-naphthalenediylcarbonyl)) resins. The second, which was granted in 2022, pertains to polylactic acid (PLA) articles that are blended with up to 25% recycled material. Chemical recycling is the only technology and supplier that is approved for this material.

Excluding an unidentified process from one of the earliest submissions in 1990 for grocery bags, all 125 of the remaining applications are physical recycling processes. The FDA has evaluated submissions with a positive outcome for a broader range of materials and applications, including both pre- and post-consumer inputs. This covers processes for flexible materials, such as linear low-density polyethylene (PCR-LLDPE), which is approved for use in 100% [132] of food contact articles for all food types restricted to specific conditions according to each company. Besides traditional polymers, in the category *others*, only polycarbonate (PC) is authorized to be used in water containers of up to 75% PCR-PC derived from water containers.

Recent years have witnessed a rise in the quantity of issued NOL. By 2024, polypropylene emerged as the leading polymer, with polyethylene ranking second and polystyrene trailing further behind, following the trend that is also observed in Europe and aligned with the volumes of plastic produced worldwide (where these two polymers also lead the list [5]).

### 2.3. EFSA vs. FDA

The procedure for obtaining authorization to use recycled plastic materials in contact with food varies between the United States (U.S.) and the European Union (EU). In the EU, the EFSA evaluates the entire recycling process, considering migration limits for specific molecules under the same regulations that apply to virgin materials [29]. A new regulation published by the European Commission in September 2022 allows the utilization of innovative technologies to generate data necessary for establishing regulatory and/or suitable processes, but it does not provide a comprehensive description of suitable methods for generating such data [133]. The FDA has established regulations governing the use of plastic materials intended to come into contact with food, with an approach based on the safety of the substances and potential contaminants rather than the entire recycling process. Data submission is encouraged by the American authority unlike the obligation required by the EFSA.

The primary distinction between the FDA and EFSA lies in their approach to the evaluation of recycling processes. While the FDA encourages the submission of information for the assessment of a recycling process’s adequacy, the EFSA has made it mandatory. The FDA provides a specific methodology and analysis to test the efficiency of a decontamination process, indicating detailed recommendations for potential surrogate materials, times, and calculations. However, the EFSA takes a more conservative approach by examining all possible aspects to verify that the process produces a food-grade material.

The European Commission requires individual authorization for each process and is strict with new technologies, which must be evaluated and approved before commercial use, including testing for functional barriers when required. To ensure complete decontamination, the EU considers both microbiological and toxicological risks. On the other hand, the American regulation allows different types of recycling (primary, secondary, and tertiary) without requiring individual authorization for each process, as long as contaminant levels are kept safe. It also allows the use of surrogates and does not require a whole decontamination but instead aims to reach secure levels of contamination.

Another differentiation lies in the terminology used to describe certain concepts. In terms of equivalence, primary recycling under FDA guidelines aligns with closed-loop controlled recycling as defined by the EFSA, where clean materials are used without requiring extensive decontamination. Secondary recycling in FDA guidelines involves the mechanical reprocessing of post-consumer plastic waste, including grinding, washing, and remelting to produce new products. This process is often considered open-loop recycling in EFSA guidelines, as materials may come from various sources, not exclusively food-grade inputs and, thus, require individual authorization due to contamination risks. Finally, tertiary recycling in FDA regulations corresponds to chemical recycling in EFSA guidelines, involving depolymerization and rigorous purification to ensure safety for food-contact applications.

The FDA employs a more flexible and recommendation-based approach, while the EFSA imposes stricter standards and specific requirements for decontamination technologies and processes prior to approval. Hence, the number of submissions with positive outcome evaluated by the FDA far surpasses that of the EFSA, with its more specific evaluation approach, which allows for a direct and focused assessment of a process for submission, while the EFSA’s general approach considers a wider range of concerns and opinions for each situation and material, enabling it to explore various implications and potential health risks associated with contaminants. Figure 3 highlights key differences between the regulatory frameworks of both regions, with factors such as migration testing, decontamination efficiency, traceability of recycled materials, and the use of functional barriers.

Concerning feedstock, the EFSA has previously established that it cannot include more than 5% of plastics from non-food consumer applications, at least for PET applications [134,135]. However, as previously presented, for non-PET materials, achieving a feedstock of at least 99% in packaging previously used in food applications was considered a critical quality factor for approval [107]. As for the FDA, it does not specify a percentage limit for non-food contact materials in the recycling stream and only focus on controls at the source.

According to EFSA challenge tests, contaminants should cover polar and non-polar compounds (with molecular weights of up to 1000 Da) [110]. Equally important is the evaluation of health risks, considering a threshold of toxicological concern (TTC) under 0.0025 μg/kg of body weight per day for unknown contaminants, which represents a negligible risk to the consumer [11]. Meanwhile, the FDA’s more specific guidance presents contaminant recommendations for a surrogate cocktail, using at least one per category according to physical and chemical properties. Volatile polar examples given are Chloroform, Chlorobenzene, 1,1,1-Trichloroethane or Diethyl ketone. Toluene is a volatile non-polar, Copper(II) 2-ethylhexanoate is a heavy metal, Benzophenone or Methyl salicylate is a non-volatile polar, and for non-volatile non-polar, Tetracosane, Lindane, Methyl stearate or Phenylcyclohexane are recommended. The guidance also states that exposure to chemical contaminant EDIs from recycled food-contact articles should be in the order of 1.5 mg/person/day (0.5 ppb dietary concentration) [117].

Regarding the test sample quantity required by both regulations, the EFSA has positively evaluated previous challenge tests conducted at pilot plant scale for PET applications [134,135]. Meanwhile, the FDA does not specify the scale of the accepted test size and specifically recommends against analyzing actual batches of post-consumer plastics. Instead, the FDA suggests using surrogate contaminant testing to evaluate the effectiveness of recycling processes, as this approach provides consistent and controlled assessments without the variability associated with changing waste compositions [117]. This method aligns with the EFSA’s approach to validating decontamination efficiency.

In relation to the final use of the approved materials, both authorities define specific conditions under which the use of the material is authorized. The EFSA, in its previous evaluations, tends to provide a detailed description of the conditions of use. For the FDA, this also depends on each specific case; some approvals indicate general conditions of use, while others specify more detailed restrictions. There are also cases where use is allowed with all types of food [131].

While both the United States and the EU face variability in recycling policies, the EU operates under a framework of common directives that establish overarching recycling targets, although implementation and enforcement vary by country. In contrast, the United States lacks a federal standard for plastic recycling, resulting in a decentralized system where recycling policies and infrastructure vary significantly across states and municipalities [52]. This disparity results in an uneven development of PS sorting and recycling strategies, with some regions implementing effective systems while others face greater challenges.

Initiatives for educating consumers on waste separation enhance recycling efficiency and foster greater environmental awareness. Additionally, factors such as convenient recycling systems, the normalization of sustainable behaviours, and the availability of resources, including information and recycling bins, contribute to more effective waste management practises [136].

### 2.4. Global Regulations

Beyond the European Union and the United States, regions like China, South Korea, the UK, and Latin America have different regulations for recycled plastics. In China, the National Health Commission (NHC) published the GB 4806.7-2023 National Food Safety Standard on Food-Contact Use Plastic Materials and Articles, which came into force in September 2024. However, this regulation did not allow the use of recycled plastics in food contact applications [137]. In November 2024, a voluntary standard, GB/T 45090-2024, titled Plastics—Identification and Marking of Recycled Plastics, was published. This standard established requirements for the labelling and marking of recycled plastics, aiming to promote uniform identification of recycled plastics. It was scheduled to take effect in June 2025 [138]. Along with a new regulation for migration, the standard GB 31604.1-2023 National Food Safety Standard on the General Rules for Migration Testing for Food-Contact Materials and Articles [137].

In South Korea, food contact materials and articles are regulated under the Food Sanitation Act, directed by the Ministry of Food and Drug Safety (MFDS). Updated standards were established through *Notice No. 2021-76: Standards and Specifications for Utensils, Containers and Packages*, which regulates the use of recycled synthetic resins, including enhanced standards for recycled polyethylene terephthalate (PET) [139].

Since the United Kingdom’s exit from the European Union in 2020, legislation on recycled plastics intended for food contact continues to be based on Regulation (EC) No 282/2008, which has been adapted into British law to maintain high food safety standards [140]. In addition, since April 2022, the UK Plastic Packaging Tax (PPT) has imposed a levy of £200 per tonne on manufacturers or importers of plastic packaging containing less than 30% recycled content, incentivizing the use of recycled materials in packaging production [141]. Moreover, the Packaging Waste Regulations Act 2023 supports the implementation of the Extended Producer Responsibility (EPR) scheme by requiring producers to report data on packaging placed on the market, ensuring accurate fee calculations and promoting sustainable packaging practises [142].

In Latin America, the Mercado Común del Sur (MERCOSUR, including Argentina, Brazil, Paraguay, and Uruguay) establishes a general criterion for plastics in food contact through *Resolution GMC No. 56/92* (General Criteria for Plastics for Food Contact), which establishes that the MERCOSUR Commission of Specialists may study special technological processes for obtaining resins from recyclable materials. Since 2024, these regulations have been under review for new requirements. The 2007 *Reglamento Técnico MERCOSUR sobre Envases de Polietilentereftalato (PET) Postconsumo Reciclado Grado Alimentario* (Technical Regulation on Recycled Post-Consumer PET for Food Contact) is highlighted [143].

## 3. State-of-the-Art Decontamination Technologies for Post-Consuming PS

Among the different types of articles in contact with food, approximately 40,000 to 100,000 substances are estimated to be present [144,145,146]. Recycling technologies must be capable of coping with the variety of substances that might be present, focusing on one material but adapting to other materials according to the properties and characteristics of each one, favouring economic viability at an industrial level.

Although polystyrene’s dense and low-permeability structure reduces the retention of unwanted substances in its matrix, the contaminants that do persist throughout its lifespan can be more challenging to remove [147].

The most common methods to clean polystyrene, aside from chemical recycling, are super-cleaning processes, which focus on removing contaminants under vacuum treatment, mainly during extrusion, combining thermal decontamination for volatiles and melt filters for non-volatiles [30]. Dissolution–precipitation is another way, where specific solvents are used to dissolve the polymer and remove contaminants to achieve food contact grade; then, an antisolvent is used to precipitate the polymer. Some examples of solvent/antisolvent systems, such as p-Cymene/heptane and limonene/xylene, have been described in the following sections, particularly in Table 2.

The dissolution–precipitation method consists of dissolving the polymer using an appropriate solvent, eliminating undissolved impurities, establishing a solid–liquid separation process, inducing polymer precipitation by introducing an antisolvent to the solution, and extracting the polymer through filtration and drying procedures [148,149].

Different researchers have explored recycled polystyrene for food contact purposes. Notable advancements have been made in the characterization of polymer properties throughout the recycling process, such as morpho-structural and thermo-mechanical [150]. However, most investigations have focused on detecting contaminants and their potential harmful migration.

Migration is influenced by various factors, primarily the nature of the packaging material and the food, the type, duration, and temperature of contact, as well as the properties and concentration of the migrating substance in the material [151]. During the decontamination phase, it is essential to consider various compounds that can migrate from food packaging. These include plasticizers, stabilizers, additives, antioxidants, and solvents. Additionally, oligomers and monomers such as styrene, which is particularly significant in polystyrene [152,153,154,155,156], are among the substances that require special attention [151].

Numerous studies concentrate on a certain type of polystyrene. For instance, the bulky nature of recycled expanded polystyrene (EPS) presents significant challenges in waste management, as it occupies substantial space in transportation and landfills while contributing to environmental pollution [155,157,158]. Primarily, researchers aim for volume reduction by studying its dissolution in different types of solvents [159,160,161,162]. However, persistent fish-related odours, mainly from trimethylamine and dimethyl sulphide, pose an additional challenge [163].

The most common research approaches include analyses of the identification of volatile organic compounds (VOCs) and semi-volatile organic compounds (SVOCs) [12,114,151,157], as well as identification and evaluation of non-intentionally added substances (NIAS) [114,158,164,165]. Non-volatile substances have not been the main focus, nor has the implementation of procedures to decontaminate any of the above-mentioned [48].

Due to their high diffusivity, polyolefins tend to exhibit significant migration, whereas polystyrene, with much lower diffusivity, shows a much slower contaminant migration under typical storage temperatures [48]. Unlike other materials, the analysis of washing and decontamination of PS is mainly studied as part of testing batteries of different or mixed plastics under common parameters [114,164,166,167]. Among these analyses, relevant results are observed for odour removal with different agents, identifying the best performance.

Polyolefins, being highly diffusive, lead to significant migration rates, whereas polystyrene exhibits much lower diffusivity, resulting in a considerably slower migration of contaminants under typical storage conditions.

Table 2 summarize the publications that have centred or mentioned a specific procedure for washing or decontamination of recycled PS, with the aim of reaching a certain food grade level. A study particularly performed on polystyrene focused on the removal of undissolved substances polybutadiene–polystyrene (PB-PS) particles and pigments. Poor retention with 1 μm membrane and low fluxes using 0.1 and 0.45 μm filters after 1 h at 20 bar achieved 100% removal of TiO_2_ and Cr/Sb/ Ti oxide was achieved after 30 min centrifugation in limonene [148].

To determine whether the solvent content in recycled pellets meets regulatory standards for commercial-grade applications, it is essential to enhance the detection threshold [168]. The effectiveness of deodorization realized by Roosen et al. [166] is influenced by diverse factors such as the ratio of solid to liquid, residence time, and the rate at which the washing solution is recirculated. Compared to the conventional water and caustic soda used in industry processes, organic solvents and detergents demonstrated superior average removal efficiency within a 15 min timeframe. Increasing agitation speed beyond 200 rpm did not enhance the removal of odours. Meanwhile, in 2022 [164], detergents (both industrial and commercial) proved to be the most efficient method for eliminating odours from polystyrene trays at 25 °C, with deodorization rates up to 58% and 67% at 65 °C.

Experiments carried out by Demets et al. [169] concluded that predominantly apolar volatile compounds can be detected in polystyrene, while less volatile components remain strongly bound to the hydrophobic polymer matrix. A general reduction of 97% in thermal desorption and 44% in solvent desorption was observed. However, reprocessing can lead to the formation or release of compounds that may impact the material’s odour.

A technology patented by the University of Alicante is highlighted, developed by the research group Engineering for the Circular Economy (I4EC), who designed a procedure for removing organic NIAS contaminants in recycled plastic materials. This process operates at an atmospheric pressure, using a non-volatile, water-soluble extraction agent. The procedure includes the stages of selection, crushing, washing, rinsing, drying, and decontamination.

In plastic recycling, detergents, caustic soda (NaOH), and water at different temperatures are commonly used as standard washing agents at the industrial level. Meanwhile, solvents like ethyl acetate have been studied and evaluated for their potential to achieve deep cleaning in polymer waste purification [164,170,171,172].

Research has shown that recycled polystyrene (PS) can be safely utilized in food contact applications when integrated into multilayer systems with functional barriers, effectively isolating potential contaminants in the inner layers [173]. During recycling, the thermal degradation of PS has been linked to an increased release of substances into vegetable oils under high temperatures. Moreover, recycled PS often contains higher levels of oxygenated styrene derivatives, such as acetophenone and benzaldehyde, while virgin PS is characterized by greater concentrations of styrene monomers and dimers [12,174,175].

Deodorization of EPS fish boxes has also been explored. These boxes are widely used for seafood storage and transportation and are usually part of a closed-loop recycling system, where proper decontamination ensures their potential reuse. Various techniques have been investigated to address this issue and improve recyclability, including acid neutralization, treatment with heated vegetable oil, and masking with limonene. Additionally, experimental approaches such as melting in organic solvents, shear compression crushing, and styrene oil production are being studied for Styrofoam waste processing, with desalting and volume reduction using heated vegetable oil showing promising potential for odour removal [163].

Appropriate challenge tests with different types of surrogate contaminants, migration models as support, or any adequate evidence must demonstrate decontamination efficiency in reducing contamination levels to a safe concentration.

**Table 2 polymers-17-00658-t002:** Decontamination methods studied at laboratory level for polystyrene decontamination.

Decontamination Method, (Author, Year)	Material	Focus	Methodology Applied
Dissolution–precipitation technique (Kol et al., 2023) [148]	HIPS	Removal of polybutadiene–polystyrene (PB-PS) particles and pigments, including TiO_2_ (white colour), Cr/Sb/Ti oxide (yellow colour), and carbon black (black colour).	Filtration with 0.1, 0.45 and 1 μm membranes, at 500 rpm constant stirring, pressures applied between 1.5 and 30 bar, 5 wt% polymer concentration in xylene and limonene, with centrifugation at 10, 30 and 60 min.
Dissolution–precipitation technique (Kara Ali et al., 2023) [168]	Recycled PS from a pilot plant	Determination of solvent content remaining in PS at various stages of a dissolution/precipitation recycling process, using p-cymene as a green solvent and heptane as an antisolvent for precipitating.	Models were made to quantify the remaining solvent content based on calibration dissolutions of 30 wt% PS in 1,2-dichloroethane (DCE) and known relative concentrations of cymene (0–16 wt%) and heptane (0–40 wt%). Samples were analyzed by Fourier Transform Infrared spectroscopy (FTIR) with ATR and deuterated L-alanine-doped tri-glycine sulphate detector.
Deodorization (Roosen et al., 2021) [166]	Plastic film waste (59.1% PE, 23.9% PP, 10.6% PET, 5.6% PVC, and 0.8 wt% PS).	Desorption isotherm and kinetic models for deodorization efficiencies in different washing media (distilled water, CTAB 0.92 mM, NaOH solution 1 wt%, a caustic soda mixture (NaOH 1 wt%,) with CTAB (0.92 mM), and ethyl acetate).	Shredded plastics (3, 4, 5, 6, 7, and 8 g) were stirred in 100 mL of each washing medium at 25 °C and 65 °C for desorption isotherm studies. Kinetic experiments (adding a 45 °C experiment for water and ethyl acetate), stirring in a shaker 5.0 ± 0.1 g of plastic at 200 rpm, removing at 0.5, 2, 4, 15, and 60 min. Vacuum filtration was used to separate the washing media, and the drying occurred at ambient temperature for 4 h.
Washing (Roosen et al., 2022 [164]	Post-consumer polystyrene (PS) trays	Deodorization in washing mediums (tap water; CTAB (9.2 mM), NaOH (2 wt%, 9.2 mM CTAB in 2 wt% NaOH solution; commercial detergent (one 18 g capsule/100 mL water) and an industrial detergent (0.5%v in 2 wt% NaOH solution), odour compounds identification, polarity chemical classes influence their removal.	Shredded 5 ± 0.1 g plastic samples were mixed with 100 mL of each medium at 25 and 65 °C and agitated using a multi-flask rotary shaker at 200 rpm for 10 min. Subsequently, they were separated by filtration, rinsed with 25 °C 100 mL distilled water and dried for 24 h in a desiccator at room temperature.
Washing (Demets et al., 2020) [169]	Flexible post-consumer film waste stream	Qualitative and semi-quantitatively techniques to analyze volatile contaminants before and after washing and pelletizing	The washing process involved rinsing, followed by a friction washer and a sink–float separation system, all using tap water. The materials were then dried with hot air (washed films) and subsequently pelletized using a vacuum-degassing extruder at 200 °C.
Dissolution–precipitation technique (Fullana et al., 2021) [176]	Mix plastic input, only tested with PE, PP, and PET.	Spanish patent technology P201931143, “Procedimiento para la descontaminación de plástico reciclado” (Procedure for the decontamination of recycled plastic). From 2019 developed at lab/pilot scale.	Separation, shredding, washing, rinsing, drying, and decontamination by means of extraction with water-soluble solvent with boiling point above 180 °C, in this case polyethylene glycol (PEG), at atmospheric pressure and subsequently rinsing. After the extraction, the plastic is rinsed at room temperature and centrifuged before and after to remove the solvent. Includes water recuperation systems by ultrafiltration or crystallization and flocculation-decantation. Meanwhile, solvent recovery by means of ultrafiltration membrane and filtering.
Deodorization (Ishida et al., 2020) [163]	Expanded polystyrene (EPS) fish boxes	Removal of fish-like and sea-like odours (trimethylamine and dimethyl sulphide) to improve recyclability.	Desalting and deodorization using heated vegetable oil (Oshima College Method—OCMT). EPS was immersed in heated vegetable oil (160–200 °C) for volume reduction and odour removal. The solubility of trimethylamine and dimethyl sulphide in vegetable oil was evaluated using Hansen solubility parameters, and desorption was experimentally tested by analyzing odour reduction in treated samples.

### Industrial Level Decontamination Technologies

Companies are involved through industrial technologies, which have been officially proposed in Europe for PS decontamination and, subsequently, for direct food application. The Styrenics Circular Solutions (SCS) association is highlighted, as it spans multiple value chains (PS, EPS, XPS, etc.) to promote the circularity of this plastic. Two technologies for polystyrene super-cleaning are highlighted under this initiative, NGR technology [70,90] and Gneuss [177]. INEOS Styrolution, a founding member of the Styrenics Circular Solutions (SCS) association, has developed technologies to treat post-consumer polystyrene as well. The evaluation of mentioned technology was presented previously to the normative change in 2022 and was consequently terminated, hence, why it did not move forward to the stages of detailed technical evaluation [88].

Technologies that were previously only used to recycle PET bottles have been expanded, opening up the possibility of also treating PS.

A technology presented for approval, that evolved from PET recycling in Europe, was the process of the association Styrenics Circular Solutions (Gneuss3) [91,92]. Gneuss technology was developed in Germany and based on an extrusion process, that has already been used in Japan, Colombia, and the USA [178] with an NOL issued by the FDA in 2009 [179].

The process is initiated with a high-pressure backflush segment through an automatic screen ultra-fine filter or Rotary Screen Filter: RSFgenius (WO/2001/043847 [180]) focusing on non-volatile impurities. Afterwards, to increase the surface area in contact with the polymer, the extruder-to-sheet patented by Gneuss for food-grade recycled post-consumer rigid and foamed polystyrene uses a multi-rotation system (MRS) with a particular screw, allowing a more efficient degassing with short residence times preventing thermal damage (WO/2003/033240 [181]). To increase the capture of volatiles in the process, the technology implements its own adapted vacuum system [177]. The company holds over 100 international patents [182] associated with technological systems and methodologies, including those focused on general methodologies such as the recycling of hydrolyzable polycondensates like polyester (PET) (WO/2020/108705A1 [183] and WO/2021/008659A1 [184]). Patents directly related to the technology described for polystyrene are cited within the text.

Assessments have been made to determine the decontamination efficiency considering the wide range of potential contaminants selection, such as high and low volatiles, polar, non-polar, etc. Challenge tests at different specific conditions are carried out, testing the suitability of the materials obtained, allowing the approval of the respective authorities in the above-mentioned countries for the use of post-consumer PS for food contact, processed in each implemented recycling plant. For example, in the Colombian case, Invima, as the authority in charge, authorized the use of the material obtained for food contact from the recycling plant of the company Alpina, and PS has already been implemented, with the introduction of 30% recycled PS in the commercial packaging of one of their products [185].

Another super-cleaning process is the Austrian NGR Technology for plastic recycling [186]. It specifically works in the melt state, uses liquid-state polycondensation (LSP) under vacuum, where the polymer increases the viscosity to remove contaminants [187]. The technology consists of four steps: (1) Conventional recycling performed by suppliers: grinding into flakes, posterior intensive washing (>70 °C, >1% NaOH, >5 min), sorting process and drying; (2) flakes remelting; (3) decontamination under melt vacuum; and (4) pelletization. Up to 100% of the recycled material is intended to be applied to dairy products, trays, or cups [188]. Studies through challenge tests have been publicly shared and conducted based on EFSA regulations but also following FDA recommendations, with details on contamination and process efficiency. Applications for authorization from the EFSA under this technology have been made successfully for PET; however, applications for polystyrene were terminated in May 2024 [69] due to changes in the regulation. Screw extruders employed to melt the polymer at elevated temperatures are an important source in generating VOC emissions from plastic due to degradation and decomposition products. Higher heating rates contribute to minimizing the time of the polymer at elevated temperatures, resulting in reduced volatile formation. Studies have also demonstrated that lower VOC content was produced in the absence of oxygen. For this reason, the vacuum extrusion process is incorporated, with the aim of removing the volatiles formed as they are generated, favoured by mass transfer [114].

In Ineos’s approach to their proposal to the EFSA for approval in 2022, they used a twin screw degassing extrusion process, similar to the SCS technology, operating under vacuum for the remotion of moisture and oxygen at high temperatures, based on PET recycling technologies already approved by the EFSA [49]. Ineos Styrolution mechanical recycling starts with a hot wash of the flakes from polystyrene bales already sorted using deep near-infrared (NIR) and object recognition. New flake sorting is implemented before proceeding to the super-cleaning technology. The material also goes through melt filtration and pelletizing [189].

Ineos already has some varieties of recycled polystyrene commercially available in the market, planned to be used as a functional barrier between two virgin layers of polymer for food contact purposes [190]. In 2023, the implementation of a recycling plant for post-consumer PS in Germany was announced, with the collaboration of the feedstock supplier, specialized in collection and sorting, Tomra, and the local recycling company, EGN. The plant is expected to start operations in 2025, with a focus on food contact applications [49,191,192].

A collaborative project with the German dairy manufacturer Unternehmensgruppe Theo Müller enabled the registration of their super-cleaning process as a novel technology. In 2024, a voluntary consumer test was conducted using yoghurt cups produced with mechanically recycled polystyrene. The results showed a positive response, with 90% of participants expressing willingness to purchase the product despite minor visual differences in the recycled packaging. These findings support the feasibility of implementing this novel technology at an industrial scale, leading to the planned market introduction of the packaging in early 2025. This initiative aims to generate the necessary data for the approval process to classify it as a suitable technology [189,193].

Despite the advances and efforts made in post-consumer polystyrene recycling over the years, there is still a potential risk of NIAS migration. Styrene is one of the most mentioned compounds to be considered, due to the impact when being exposed to it, such as irritation of the skin, eyes, respiratory tract, and depression of the central nervous system. With an average of 100 to 3000 ppm of styrene detected in food packaging, its oxidation to styrene oxide can also cause health problems [151,156]. The extensive list of potential contaminants that can be present in a recycled material requires an exchange of information between different areas to address the uncertainties of today.

There are no defined methods for risk assessment that combine the various factors involved. The presence of unintended substances for food contact, which may come from labels such as inks, cleaning products from both the container and its contents, related with inappropriate use, or improper classification at the entry of the process, and even from other unknown products that are usually within the consumer’s requirements, such as transparency in the information about their composition. Thus, this is the problem that various technologies face when decontaminating a material. Therefore, multidisciplinary work should enable experts from different fields to work together to evaluate potential reactions and mixtures that may occur throughout the process, which in turn could represent a health risk if they remain until the end of the process.

## 4. Conclusions

The regulatory landscape for post-consumer recycled (PCR) plastics in food contact applications continues to evolve, with the European Union (EU) and the United States (U.S.) adopting distinct approaches. The European Food Safety Authority (EFSA) enforces a rigorous process that mandates the evaluation and approval of each recycling technology, ensuring comprehensive decontamination and risk mitigation across the entire value chain. In contrast, the U.S. Food and Drug Administration (FDA) follows a more flexible framework, providing recommendations rather than mandatory approvals, placing the responsibility on manufacturers to demonstrate product safety.

Both regulatory bodies prioritize the control of contaminant migration, but the EFSA’s stringent requirements lead to a more structured risk assessment, whereas the FDA permits greater adaptability. The case of post-consumer recycled polyethylene terephthalate (PCR-PET) exemplifies how a well-established decontamination process, combined with low-diffusion properties, facilitates regulatory acceptance. However, expanding food-contact applications to other polymers presents significant challenges. While polyolefins (PE and PP) face obstacles due to high diffusivity, polystyrene (PS) emerges as a promising candidate given its low permeability and thermal stability. Yet, its approval hinges on the development of advanced decontamination technologies capable of effectively removing potential contaminants, including non-intentionally added substances (NIAS).

The presence of NIAS remains a critical issue, particularly in PS and polyolefins, where the migration of monomers and degradation products, such as styrene oxide, raises health concerns. To address these risks, regulatory agencies advocate for extensive challenge testing and improved analytical methods to assess and control NIAS levels in recycled plastics. Recent advancements in decontamination technologies, including super-cleaning and dissolution–precipitation processes, demonstrate potential for achieving the necessary purity standards. Notable industrial efforts, such as those by Styrenics Circular Solutions, NGR, Gneuss, and Ineos Styrolution, have made progress in enhancing PS recycling technologies. However, regulatory approval remains an ongoing process, requiring further optimization and validation.

Ultimately, the successful integration of recycled plastics into food-contact applications depends on three key factors: robust regulatory frameworks, continuous technological innovation, and interdisciplinary collaboration. Aligning global regulatory standards and refining decontamination techniques will be essential to expanding the safe use of PCR materials. Overcoming these challenges will not only facilitate the approval of additional recycled polymers for food contact but also contribute to the broader goal of a sustainable, circular plastics economy.

## Figures and Tables

**Figure 1 polymers-17-00658-f001:**
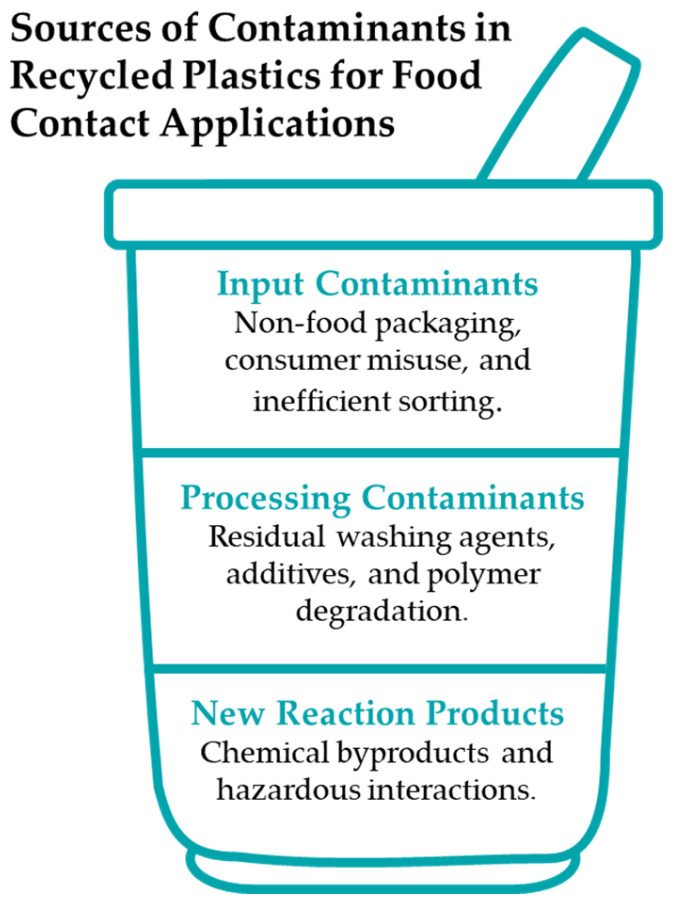
Schematic representation of contaminant sources in recycling processes.

**Figure 2 polymers-17-00658-f002:**
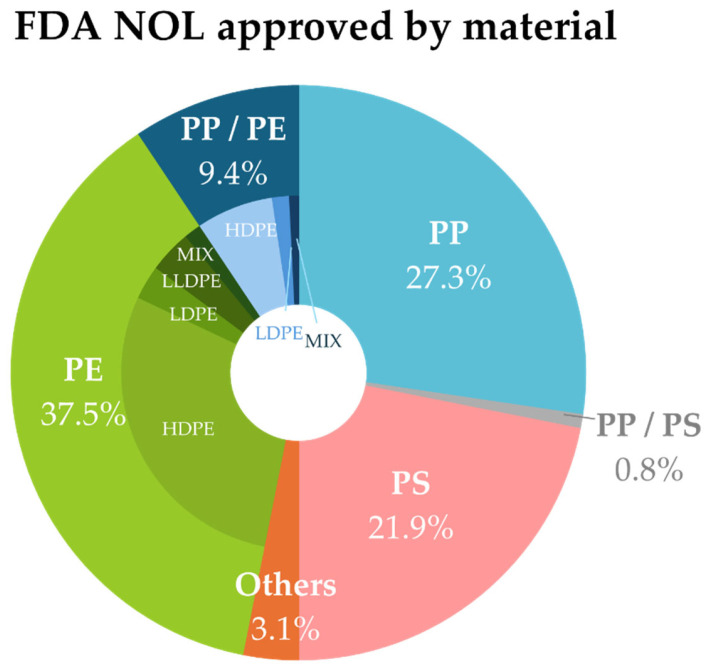
Distribution of favourable recycling process submissions with no objection letter (NOL) granted by the FDA on post-consumer recycled (PCR) non-PET plastics intended for food contact applications classified by material type (1990–2024) [131].

**Figure 3 polymers-17-00658-f003:**
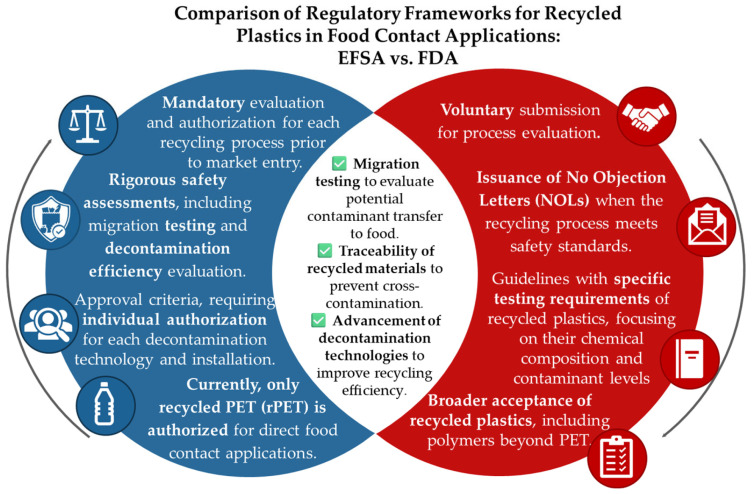
Comparison of regulatory frameworks for recycled plastics in food contact applications in the EU (**left**) versus in the United States (**right**).

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
