# Peer review of "Regulatory Frameworks and State-of-the-Art Decontamination Technologies for Recycled Polystyrene for Food Contact Applications"

_polymers, 2025, doi:10.3390/polym17050658_

Round 1
Reviewer 1 Report
Comments and Suggestions for Authors
This is a review manuscript on the non-PET materials and polystyrene recycling methods and legislation for food contact applications. This is a comprehensive review covering all aspects of the recycling and also European and non European and FDA legislation. It is an interesting review which can be helpful and useful for experts and industries, however, it needs restructuring and also deleting some repeated texts. Also, title of the manuscript is about polystyrene, but all packaging materials were discussed, so please change and revise the title.
It would be interesting to have a figure showing the different packaging materials and their application and rate of their recycling.
It would be better to shorten the text and written materials, one way could be to present them as figure, which could make this manuscript easy to follow for readers. For example Lines 80-84 could be as figure. Also draw a flowchart showing the different administration and their requirements in this field. OR a figure for different recycling systems with their advantage and drawbacks.
Delete some unrelated texts as this manuscript is so long, for example lines 116 -130, or lines 164-170.
Like manuscript itself, the conclusion section is also so long. It should be in just one paragraph.
Author Response
Comment: This is a review manuscript on the non-PET materials and polystyrene recycling methods and legislation for food contact applications. This is a comprehensive review covering all aspects of the recycling and also European and non European and FDA legislation. It is an interesting review which can be helpful and useful for experts and industries, however, it needs restructuring and also deleting some repeated texts. Also, title of the manuscript is about polystyrene, but all packaging materials were discussed, so please change and revise the title.
Response: We Thank Reviewer for his/her constructive feedback and for accepting our manuscript for its publication in Polymers. The manuscript has been revised for improved clarity and coherence. Although same other plastics have been mentioned, the title references polystyrene (PS) as it is the main focus of the current manuscript, where detailed discussions on other materials are not included but only mentioned for comparison. In fact, these were only mentioned in earlier sections to provide context on the regulatory framework and how different regions assess recycled food-contact materials.
Comment: It would be interesting to have a figure showing the different packaging materials and their application and rate of their recycling.
Response: Thank you for your suggestion. While a figure showing the different packaging materials, their applications, and recycling rates would be interesting, it was not included in this review. Instead, we prioritized the inclusion of the other figures proposed later in the manuscript, as they are more specific to the focus of this review.
Comment: It would be better to shorten the text and written materials, one way could be to present them as figure, which could make this manuscript easy to follow for readers. For example Lines 80-84 could be as figure. Also draw a flowchart showing the different administration and their requirements in this field. OR a figure for different recycling systems with their advantage and drawbacks.
Response: Following your suggestions, Figures 1 and 2 have been incorporated to present relevant information in a clearer format, corresponding to lines 86 and 738, respectively. Nevertheless, it should be mentioned that the manuscript although could appear so long for the reviewer, it has the typical extension of a review paper.
Comment: Delete some unrelated texts as this manuscript is so long, for example lines 116 -130, or lines 164-170.
Response: The text has been reduced by removing redundant content, but not deleted. However, although we appreciate the reviewer comments, we do not agree with the text is unrelated and it could not be removed because it support the response to another reviewer who highlighted how we rightly described the biological degradation of polymers as it is another end-life option and asked us to extend the manuscript including thermal recovery (incineration). This part has been added to the current version of the manuscript, and we believe that the manuscript has now improved with that information.
Comment: Like manuscript itself, the conclusion section is also so long. It should be in just one paragraph.
Response: The conclusion has been condensed.
Reviewer 2 Report
Comments and Suggestions for Authors
Dear Authors,
Your publication provides an excellent and comprehensive summary of the legal framework surrounding the recycling of polymers and their use as food packaging in Europe and the USA. Particularly noteworthy is the well-structured compilation of current findings on polystyrene.
However, I would like to address a few points that require further clarification:
- Formatting: The page numbering ends at page 8, followed by a table, after which the numbering appears to restart at page 1. Please correct this issue.
- Line 109 ff.: The authors mention the possibility of the biological degradation of polymers and rightly describe it as a waste reduction strategy. However, I miss the inclusion of thermal recovery (incineration). This method is also a waste reduction strategy but has the added advantage of energy recovery. Wouldn't this approach be preferable to biological degradation, especially considering the potential issues of micro- and nanoplastic release into the environment? Including this aspect would enhance the completeness of the discussion. What is the authors' stance on this matter?
- Line 242 ff.: To clarify the argument, the EU plastic tax could be mentioned here. A fee of 800 euros per ton applies to single-use plastic packaging.
- Line 367: The term in question has already been spelled out and abbreviated in line 55. Please limit its full definition to one occurrence per publication.
- Line 611 ff.: The content of this section is already covered in lines 599 ff. Please remove the duplicate information.
- Line 647: The notation "3°" is unclear. Should this be "3rd"? Please adjust accordingly.
- Lines 743 & 763: It would be helpful to include examples of the solvents used, providing readers with direct references and reducing the need for external sources.
- Line 774: What about EPS fish boxes used for the cooling and storage of seafood at fish markets? These pose significant challenges in terms of deodorization.
- Line 777 ff.: The content of this section is already addressed in lines 770 ff. Please remove the redundant passage.
Additionally, I have a further question related to the publication: To what extent do drinking water pipes fall under the regulations mentioned? While they are not traditional packaging, they do transport food products. Could this be a valuable topic for further research?
Author Response
Dear Authors,
Comment: Your publication provides an excellent and comprehensive summary of the legal framework surrounding the recycling of polymers and their use as food packaging in Europe and the USA. Particularly noteworthy is the well-structured compilation of current findings on polystyrene.
Response: We thank Reviewer for his/her valuable comments and for accepting our manuscript for its publication in Polymers
However, I would like to address a few points that require further clarification:
Comment: Formatting: The page numbering ends at page 8, followed by a table, after which the numbering appears to restart at page 1. Please correct this issue.
Response: Thank you. The formatting issue with the page numbering has been corrected.
Comment: Line 109 ff.: The authors mention the possibility of the biological degradation of polymers and rightly describe it as a waste reduction strategy. However, I miss the inclusion of thermal recovery (incineration). This method is also a waste reduction strategy but has the added advantage of energy recovery. Wouldn't this approach be preferable to biological degradation, especially considering the potential issues of micro- and nanoplastic release into the environment? Including this aspect would enhance the completeness of the discussion. What is the authors' stance on this matter?
Response: Thank you for your insightful comment. The discussion on incineration as a waste reduction strategy has been incorporated in lines 106-125, emphasizing its role in energy. Additionally, a comparison with biological degradation, highlighting that while incineration mitigates microplastic release, it also generates greenhouse gases and toxic byproducts. This broader perspective ensures a more comprehensive discussion on waste management strategies for polymers.
Comment: Line 242 ff.: To clarify the argument, the EU plastic tax could be mentioned here. A fee of 800 euros per ton applies to single-use plastic packaging.
Response: Thank you for your suggestion. A reference to the EU plastic tax has been added in lines 807-816 to clarify its role in promoting recycling and reducing environmental impact. This inclusion enhances the discussion by providing additional context on financial mechanisms aimed at incentivizing responsible plastic waste management.
Comment: Line 367: The term in question has already been spelled out and abbreviated in line 55. Please limit its full definition to one occurrence per publication.
Response: Thank you for your observation. The full definition has been limited to a single occurrence in the manuscript to maintain conciseness and avoid redundancy.
Comment: Line 611 ff.: The content of this section is already covered in lines 599 ff. Please remove the duplicate information.
Response: The duplicated information has been removed to improve clarity and avoid redundancy in the manuscript.
Comment: Line 647: The notation "3°" is unclear. Should this be "3rd"? Please adjust accordingly.
Response: Thank you for your comment. The nomenclature has been applied in accordance with the FDA's "Guidance for Industry - Use of Recycled Plastics in Food Packaging: Chemistry Considerations". However, the terminology has been modified accordingly for improved clarity.
Comment: Lines 743 & 763: It would be helpful to include examples of the solvents used, providing readers with direct references and reducing the need for external sources.
Response: In the first case, line 743, examples discussed further in the text were mentioned. Meanwhile, for the second suggestion, line 763, while providing examples of the solvents used could offer direct references for readers, this section discusses a broad and diverse range of potential migrating substances. Listing specific examples here would significantly extend the text. Instead, solvent examples are detailed in later sections to maintain clarity and conciseness
Comment: Line 774: What about EPS fish boxes used for the cooling and storage of seafood at fish markets? These pose significant challenges in terms of deodorization.
Response: Thank you for raising this important point. Expanded polystyrene (EPS) fish boxes pose a significant challenge due to their high surface area, absorption of strong odors, and contamination with organic residues. We have incorporated a discussion on the difficulties of deodorizing EPS fish boxes and how current decontamination technologies address these issues, that can be found between lines 865-866 and 925 -933, also incorporated a study about it in table 2.
Comment: Line 777 ff.: The content of this section is already addressed in lines 770 ff. Please remove the redundant passage.
Response: Thank you for your observation. The redundant passage has been removed to enhance clarity and avoid repetition in the manuscript.
Comment: Additionally, I have a further question related to the publication: To what extent do drinking water pipes fall under the regulations mentioned? While they are not traditional packaging, they do transport food products. Could this be a valuable topic for further research?
Response: As materials in contact with food, both topics share fundamental safety principles but fall under distinct regulations. While food contact materials are regulated for migration into food, drinking water pipes are subject to standards addressing long-term exposure and continuous water flow. In the EU, Directive (EU) 2020/2184 sets requirements for materials in contact with drinking water, though it applies to the entire distribution system rather than pipes alone. In the U.S., NSF/ANSI 61, a widely adopted but non-federal standard, regulates the release of substances from drinking water system components. Despite these differences, both frameworks aim to prevent harmful substance migration. Thus, while this review does not cover the topic, indeed it remains a valuable subject for further research which could lead to collaborations between researchers in both fields, the comment is appreciated.
Reviewer 3 Report
Comments and Suggestions for Authors
The review manuscript by Sepulveda-Carter et al. is very well written, with an in-depth discussion of regulatory framework between Europe and U.S. I have some detailed comments below for some suggestions and areas to improve.
0) Title and Abstract:
- It is mentioned in the abstract that European and U.S. legislation will be reviewed but the title does not specify it. I would suggest maybe including in the title.
Line 17: U.S. needs to be spelled United States. What about other countries such as China? Or South America/Mercosur region?
Line 25: Why discuss applications for HDPE and PP if the manuscript is discussing polystyrene? Also, why not include some details on the U.S. submissions (ratio of submission to polymer).
1) Introduction:
Lines 102-109: Chemical/Advanced recycling is not considered as a long-term solution because it is already available in the market. For example, LyondellBasell or ExxonMobil already sell products in the category. Many brandowners are more open to chemical recycling due to having a “virgin” plastic resin in more contact-sensitive applications, such as food packaging. Another commercial alternative in the chemical recycling space is the use of bio-based feedstock, such as used cooking oil, to produce plastics. Neste is an example of company that process it and sell it to plastic manufacturers.
Lines 191-198: I think it is important to highlight the challenges to sort PS from other plastic streams. In the United States, just recently that PP started to be sorted from other plastics, as previously only PET and HDPE (natural/milk jugs and colored) were sorted by Material Recovery Facilities (MRFs). Also, there might be some issue in educating consumers that PS can be recycled but expanded PS (Styrofoam) could not.
2) Legal framework
Lines 586-593: It might be interesting adding the categories of substances of the cocktail (e.g., volatile polar, volatile non-polar, etc.). Another comment here is in regard to the size of “batch” use to demonstrate the process. In what scale (lab scale vs. pilot scale vs. commercial scale) does the process need to be demonstrate to the FDA? It could be very challenging and costly to a large batch of “contaminated” plastic resin to simulate the process.
Lines 600-605 and Lines 611-615: It is the same text in these two paragraphs. Please review it.
Line 635: “Figure 1” with uppercase on “Figure”.
Line 663: “polypropylene” is lower case, and “Polyethylene” is with upper case. Please review it.
Section 2.3.: I miss in this section some discussion regarding source control of feedstock and foot type. Is there any difference in these areas between FDA and EFSA. Also, I understand that the goal here is to compare EFSA and FDA and the manuscript is discussing food contact, but it would be interest to mention about Cospatox guidelines. It is a protocol developed in Europe for recycled plastics in cosmetics that can be even more “sophisticated” than EFSA/FDA requirements and many cosmetics brandowners also require food-contact recycled plastics for some of their applications.
Section 2: I would suggest including one paragraph here in section 2 to briefly discuss legislation in other countries that are not US/Europe.
3) State of the art of decontamination technologies for post-consuming PS
Table 2: I’m not sure that the two studies at the end of the table refer to PS. Maybe a more recent study from Sanchez-Rivera et al. in the dissolution-precipitation method that included PS can be found here: https://doi.org/10.1016/j.wasman.2025.01.022
Line 841: Should “NLO” be “LNO”(letter of non-objection) or “NOL” (Non-objection letter)?
Section 3.1: I miss in this section some discussion regarding the washing part of the recycling. How the technologies compared across different known manufacturers (e.g., Sorema, Krones, Lindner, Amut, etc.)? The manufacturers mentioned (e.g., Gneuss, NGR) only work on the extrusion/deodorization part of the process.
Author Response
Comment: The review manuscript by Sepulveda-Carter et al. is very well written, with an in-depth discussion of regulatory framework between Europe and U.S. I have some detailed comments below for some suggestions and areas to improve.
Comment: Title and Abstract:
- It is mentioned in the abstract that European and U.S. legislation will be reviewed but the title does not specify it. I would suggest maybe including in the title.
Response: Thank you for your suggestion. The title has been maintained as it is to keep it concise and focused on the core subject of the manuscript. While the abstract explicitly mentions the review of European and U.S. legislation, the title aims to broadly encompass regulatory frameworks without specifying regions, ensuring a more general and inclusive scope.
Comment: Line 17: U.S. needs to be spelled United States. What about other countries such as China? Or South America/Mercosur region?
Response: Thank you for your comment. 'U.S.' has been replaced with 'United States' for consistency where appropriate.
Given the global influence of food packaging regulations, this review focuses on the frameworks established in the European Union and the United States, as both regions play a key role in shaping international safety standards and trade requirements. Their extensive regulatory systems set a reference for many other markets, making them particularly relevant for discussion. While other major economies also have important regulations, a brief description of legislation in other countries was included, in Section 2 (lines 789–824), with emphasize in the regions of China, South Korea, the U.K., and the Mercosur region.
Comment: Line 25: Why discuss applications for HDPE and PP if the manuscript is discussing polystyrene? Also, why not include some details on the U.S. submissions (ratio of submission to polymer).
Response: To describe the regulations of both regions in depth, the cases of HDPE and PP are examined. In European regulations, relevant information on the assessment of applications is often derived from responses to previously evaluated cases, which later serve as the foundation for official guidelines, as was the case with PET. Since no publicly available evaluations of PS applications have been published to date, specific regulatory guidance for this material remains unavailable.
Respecting the details on U.S. submissions, including the ratio of submissions by polymer type, these are summarized in Figure 2 for clarity and ease of interpretation. This visual representation provides a more concise presentation of the data while ensuring readability in the text.
1) Introduction:
Comment: Lines 102-109: Chemical/Advanced recycling is not considered as a long-term solution because it is already available in the market. For example, LyondellBasell or ExxonMobil already sell products in the category. Many brandowners are more open to chemical recycling due to having a “virgin” plastic resin in more contact-sensitive applications, such as food packaging. Another commercial alternative in the chemical recycling space is the use of bio-based feedstock, such as used cooking oil, to produce plastics. Neste is an example of company that process it and sell it to plastic manufacturers.
Response: Thank you for your suggestion. The information on chemical recycling, including the use of alternative feedstocks like used cooking oil, has been added between lines 741 and 747. The text has been revised for greater precision, and the inclusion of bio-based feedstocks has been incorporated as a complement to the paragraph. This highlights how such methods help reduce dependence on fossil resources, while plastic waste management still relies largely on landfill accumulation and incineration with energy recovery. This addition also addresses a suggestion from another reviewer.
Comment: Lines 191-198: I think it is important to highlight the challenges to sort PS from other plastic streams. In the United States, just recently that PP started to be sorted from other plastics, as previously only PET and HDPE (natural/milk jugs and colored) were sorted by Material Recovery Facilities (MRFs). Also, there might be some issue in educating consumers that PS can be recycled but expanded PS (Styrofoam) could not.
Response: Thank you for your comment. The challenges of sorting polystyrene (PS) from other plastics have been added in lines 204 -215 and 781-787.
Comment: 2) Legal framework
Comment: Lines 586-593: It might be interesting adding the categories of substances of the cocktail (e.g., volatile polar, volatile non-polar, etc.). Another comment here is in regard to the size of “batch” use to demonstrate the process. In what scale (lab scale vs. pilot scale vs. commercial scale) does the process need to be demonstrate to the FDA? It could be very challenging and costly to a large batch of “contaminated” plastic resin to simulate the process.
Response: Thank you for your suggestion. The categories of substances in the cocktail are described in lines 755–759.
Regarding the batch size for demonstrating the process, since the information is not specific about it, this aspect was discussed in detail between lines 762 and 769.
Comment: Lines 600-605 and Lines 611-615: It is the same text in these two paragraphs. Please review it.
Response: Thank you for your observation. The redundant has been removed to improve clarity and avoid repetition
Comment: Line 635: “Figure 1” with uppercase on “Figure”.
Response: The text has been corrected to use uppercase 'Figure'.
Comment: Line 663: “polypropylene” is lower case, and “Polyethylene” is with upper case. Please review it.
Response: The text has been revised to ensure consistency.
Comment: Section 2.3.: I miss in this section some discussion regarding source control of feedstock and foot type. Is there any difference in these areas between FDA and EFSA. Also, I understand that the goal here is to compare EFSA and FDA and the manuscript is discussing food contact, but it would be interest to mention about Cospatox guidelines. It is a protocol developed in Europe for recycled plastics in cosmetics that can be even more “sophisticated” than EFSA/FDA requirements and many cosmetics brandowners also require food-contact recycled plastics for some of their applications.
Response: A discussion about the feedstock and food type differences in the evaluation for both authorities has been added in lines 742-748 and 770-775 respectively.
We appreciate your suggestion regarding the Cospatox guidelines. This is indeed an important and interesting topic, closely related to the discussion on food-contact recycled plastics, and it could certainly help shape the approach to regulatory assessments. However, addressing regulations for cosmetics would significantly broaden the scope of this manuscript, which is specifically focused on EFSA and FDA requirements for food-contact materials.
Comment: Section 2: I would suggest including one paragraph here in section 2 to briefly discuss legislation in other countries that are not US/Europe.
Response: Thank you for your suggestion. We have included a brief description of legislation in other countries beyond the U.S. and Europe in Section 2, specifically in lines 789–824. This paragraph summarizes regulations in China, the UK, Japan, and Mercosur regarding recycled plastics in food contact applications.
Comment: 3) State of the art of decontamination technologies for post-consuming PS
Table 2: I’m not sure that the two studies at the end of the table refer to PS. Maybe a more recent study from Sanchez-Rivera et al. in the dissolution-precipitation method that included PS can be found here: https://doi.org/10.1016/j.wasman.2025.01.022
Response: Thank you for your observation. The studies included in Table 2 were specifically selected based on their focus on food packaging applications. While more recent studies, such as the one by Sánchez-Rivera et al., provide valuable insights into the dissolution-precipitation method for PS, our selection criteria prioritized research directly related to food-contact packaging to ensure relevance to the scope of this study. Nevertheless, this references has been added in another part between 841 and 843 lines
Comment: Line 841: Should “NLO” be “LNO”(letter of non-objection) or “NOL” (Non-objection letter)?
Response: Although the term 'Letter of No Objection (LNO)' is commonly used in this context, the FDA's official website refers to it as a 'No Objection Letter (NOL)'. The term has been standardized in the document.
Comment: Section 3.1: I miss in this section some discussion regarding the washing part of the recycling. How the technologies compared across different known manufacturers (e.g., Sorema, Krones, Lindner, Amut, etc.)? The manufacturers mentioned (e.g., Gneuss, NGR) only work on the extrusion/deodorization part of the process.
Response: Thank you for your valuable feedback. The focus of this section was primarily on decontamination rather than washing, as washing processes are already well standardized across the industry. While technologies from manufacturers such as Sorema, Krones, Lindner, and Amut play a crucial role in pre-cleaning and washing, the study primarily aimed to discuss decontamination strategies, which involve additional steps beyond conventional washing. Nevertheless, the available information on washing technologies was considered and integrated where relevant along the manuscript.
Round 2
Reviewer 3 Report
Comments and Suggestions for Authors
Thank you for proving an updated version of the manuscript. It has significantly improved after the revision. The authors have addressed most of my comments and suggestions to improve. Just some minor comments:
Line 641: There is an extra period before "To address this issue...".
Figure 3: Nice figure. In the figure title, there is "EU vs. FDA" - I would suggest having "EFSA vs. FDA" or "EU vs. U.S.".
Line 759: "In the American country" - I understand that the authors want to avoid saying U.S. or United States multiple times, but "American country" is very uncommon and I suggest finding another term here or just remove it and start the sentence with something such as: "The guidance also states that the exposure....".
Line 960: "NLO".... "NOL".
Lines 964-965: I think it is interesting to cite/reference the Gneuss patent here.
Line 979: Extra space between "technology" and "for".
Author Response
Comment: Thank you for proving an updated version of the manuscript. It has significantly improved after the revision. The authors have addressed most of my comments and suggestions to improve. Just some minor comments:
Response: Thank you for your valuable comments
Comment: Line 641: There is an extra period before "To address this issue...".
Response: The extra period before "To address this issue..." has been removed.
Comment: Figure 3: Nice figure. In the figure title, there is "EU vs. FDA" - I would suggest having "EFSA vs. FDA" or "EU vs. U.S.".
Response: Thank you for your observation. We have revised the figure title in Figure 3, changing the title to "EFSA vs. FDA" to better reflect the appropriate regulatory comparison (line 737).
Comment: Line 759: "In the American country" - I understand that the authors want to avoid saying U.S. or United States multiple times, but "American country" is very uncommon and I suggest finding another term here or just remove it and start the sentence with something such as: "The guidance also states that the exposure....".
Response: Thank you for your suggestion. We have made the recommended change by starting the sentence with 'The guidance also states that the exposure...' as reflected in line 758-759.
Comment: Line 960: "NLO".... "NOL".
Response: "NLO" has been corrected to "NOL".
Comment: Lines 964-965: I think it is interesting to cite/reference the Gneuss patent here.
Response: Thank you for your suggestion. We have added references to the most relevant patents in lines 962-973. It is important to note that there is no clear information linking each technology described by the company to a specific patent, but we have included the references that best align with the content discussed.
Comment: Line 979: Extra space between "technology" and "for".
Response: The extra space between "technology" and "for" has been deleted.